# ON DISCOVERING ALGORITHMS FOR ADVERSARIAL IMITATION LEARNING

**Shashank Reddy Chirra**$^{\diamond\dagger}$
University of Oxford
shashank@robots.ox.ac.uk

**Jayden Teoh**
Singapore Management University
jxteoh.2023@smu.edu.sg

**Praveen Paruchuri**
IIIT, Hyderabad
praveen@iiit.ac.in

**Pradeep Varakantham**
Singapore Management University
pradeepv@smu.edu.sg

## ABSTRACT

Adversarial Imitation Learning (AIL) methods, while effective in settings with limited expert demonstrations, are often considered unstable. These approaches typically decompose into two components: Density Ratio (DR) estimation $\frac{\rho_E}{\rho_\pi}$, where a discriminator estimates the relative occupancy of state-action pairs under the policy versus the expert; and Reward Assignment (RA), where this ratio is transformed into a reward signal used to train the policy. While significant research has focused on improving density estimation, the role of reward assignment in influencing training dynamics and final policy performance has been largely overlooked. RA functions in AIL are typically derived from divergence minimization objectives, relying heavily on human design and ingenuity. In this work, we take a different approach: we investigate the discovery of data-driven RA functions, i.e, based directly on the performance of the resulting imitation policy. To this end, we leverage an LLM-guided evolutionary framework that efficiently explores the space of RA functions, yielding *Discovered Adversarial Imitation Learning* (DAIL), the first meta-learnt AIL algorithm. Remarkably, DAIL generalises across unseen environments and policy optimization algorithms, outperforming the current state-of-the-art of *human-designed* baselines. Finally, we analyse why DAIL leads to more stable training, offering novel insights into the role of RA functions in the stability of AIL.

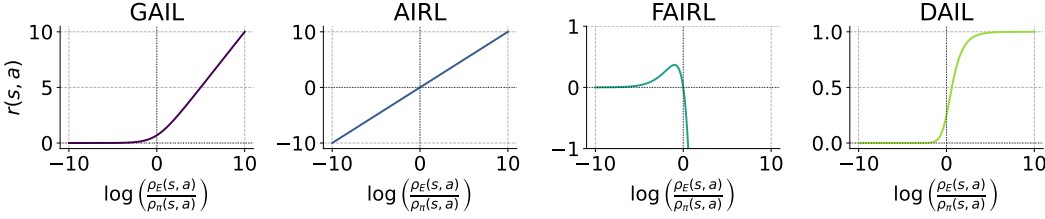

Figure 1: Visualization of the different reward assignment functions.

## 1 INTRODUCTION

*Reinforcement Learning (RL)* has achieved impressive results across a range of complex domains (Mnih et al., 2015; Silver et al., 2016), conditioned on the availability of well-defined and informative reward functions. However, in many real-world settings, specifying such reward functions

---

$^{\diamond}$ Corresponding author
$^{\dagger}$ Research conducted while at SMU as a Research Engineer

is either prohibitively difficult or entirely infeasible, whereas providing demonstrations of the desired behavior is often easier and cost-effective. This motivates the paradigm of *Imitation Learning* (IL; Argall et al. (2009); Schaal (1999)), which seeks to learn policies directly from expert demonstrations. IL is particularly well-suited for applications such as autonomous driving (Pomerleau, 1988) and robotic manipulation (Argall et al., 2009), where hand-crafting precise reward functions poses a significant challenge.

A particularly effective approach within imitation learning is *Adversarial Imitation Learning* (AIL; Ho & Ermon (2016)), which draws inspiration from *Generative Adversarial Networks* (GANs; Goodfellow et al. (2014)) and is recognized for its strong performance when expert demonstrations are limited. Similar to GANs, AIL formulates the learning process as a two-player adversarial game between a generator (i.e., the *policy* network) and a *discriminator* network. The policy aims to generate trajectories that are indistinguishable from those of the expert, while the discriminator learns to distinguish between expert- and policy-generated trajectories.

Since it shares a similar objective to GANs, AIL inherits some of the training challenges associated with adversarial methods (Arjovsky et al., 2017)—most notably the issues related to instability. In this work, we focus on one critical factor that underpins stable and effective training: **the quality of the learning signal**. Previous research in GANs (Goodfellow et al., 2014; Arjovsky et al., 2017) has shown that providing strong and informative gradient signals to the generator is essential for improving its performance and ensuring convergence of the adversarial game. In the context of AIL, this learning signal manifests as the rewards given to states-action pairs visited by the policy. Although recent non-adversarial approaches (e.g., Kostrikov et al. (2020) and Garg et al. (2021)) have shown promise, their empirical performance remains mixed (Jain et al., 2024; Lai et al., 2024), and they offer limited flexibility (e.g. for reward shaping (Sapora et al., 2024)), motivating the need to stabilize adversarial imitation learning through more informative reward signals for policy optimization.

We first introduce AIL and reward assignment through the lens of *divergence minimization* between expert and policy occupancy measures as highlighted in Ghasemipour et al. (2020). This perspective reveals a natural two-stage decomposition of the reward assignment process: (a) *Density Ratio (DR)* estimation, where the ratio of occupancy measures for each state-action pair is estimated, and (b) *Reward Assignment (RA)*, which maps this ratio to scalar rewards for policy optimization. While prior work has improved stage (a) through better stabilization of the discriminator training (Luo et al., 2024; Wang et al., 2024; Lai et al., 2024) (b) has attracted considerably less attention (Fu et al., 2018; Ghasemipour et al., 2020). Following Ghasemipour et al. (2020), we highlight how RA functions (Figure 1) influence policy-learning dynamics in adversarial training, and propose an LLM-based meta learning framework to **discover reward assignment functions** for improved performance.

This optimization produces a RA function that, when integrated into the AIL framework, results in *Discovered Adversarial Imitation Learning* (DAIL) (Figure 1). When evaluated on unseen environments from the Brax (Freeman et al., 2021) and Minatar (Young & Tian, 2019) suites, DAIL outperforms state-of-the-art baselines, including GAIL (Ho & Ermon, 2016), AIRL (Fu et al., 2018), FAIRL (Ghasemipour et al., 2020) and GAIL-heuristic (Orsini et al., 2021). To the best of our knowledge, DAIL is the first meta-learned AIL algorithm. We further demonstrate that it also generalizes to policy optimization algorithms not seen during discovery. Finally, by examining DAIL's training dynamics, we show that DAIL enhances performance by producing more informative learning signals.

## 2 RELATED WORK

**Learning from Demonstrations**   In settings where reward design is challenging or limited in expressivity (Argall et al., 2009; Teoh et al., 2025), or where exploration is difficult (Nair et al., 2018), learning directly from expert demonstrations offers a compelling alternative. The simplest approach, *Behavior Cloning* (BC; Pomerleau (1988); Torabi et al. (2018)), treats imitation as supervised learning but struggles in low-data regimes due to compounding errors outside the training distribution. In contrast, *distribution-matching* methods explicitly align the expert and policy state-action distributions, which helps mitigate distributional shift and improves robustness. Among them, *Adversarial Imitation Learning* (Ho & Ermon, 2016; Fu et al., 2018; Ghasemipour et al., 2020; Orsini et al.,

2021)—inspired by GANs—has shown strong results but suffers from instability. Previous work has focused on stabilizing the discriminator using improved loss functions (Luo et al., 2024; Wang et al., 2024), architectures (Lai et al., 2024), and regularizers (Orsini et al., 2021; Xiao et al., 2019). However, *reward assignment*—how discriminator logits should be mapped to rewards—has received little attention. Fu et al. (2018) first highlighted the impact of RA, showing that performance can vary with the structure of the underlying MDP, and proposed methods such as incorporating absorbing states to mitigate this dependence. Ghasemipour et al. (2020) and Zhang et al. (2020) highlighted the impact of RA functions on policy optimization and along with Ke et al. (2021) unified various RA functions as instances of divergence minimization within the distribution-matching IL framework, while Orsini et al. (2021) empirically benchmark these functions across multiple tasks. This work is, to the best of our knowledge, the first that explores the discovery of novel reward assignment functions, as an alternative to relying on the existing human-derived ones. Recent non-adversarial distribution-matching IL methods, such as ValueDICE (Kostrikov et al., 2020) and IQ-Learn (Garg et al., 2021), have demonstrated promise but showed mixed empirical performance, with AIL variants often performing comparably or better across multiple benchmarks (Lai et al., 2024; Jain et al., 2024). Additionally, unlike AIL approaches, they provide limited flexibility in accommodating scenarios such as state-only demonstrations (Torabi et al., 2018; Jain et al., 2024) and reward shaping (Sapora et al., 2024). Another closely related area is *Inverse Reinforcement Learning* (Ng & Russell, 2000), which seeks to infer the underlying reward function that best explains the expert policy (Skalse & Abate, 2023; Chirra et al., 2024). This contrasts with our narrower focus on directly optimizing imitation policies and designing reward assignment functions that ensure their stable training. On the other hand, works such as (Ma et al., 2024a;b; Li et al., 2025) directly tackle the problem of reward design in MDPs by leveraging LLM-based evolution, ultimately outperforming human-designed rewards.

**Meta Learning** Meta-Learning (also known as 'learning to learn') aims to automatically discover learning algorithms through end-to-end optimization (Schmidhuber, 1987; Thrun & Pratt, 1998) and has seen extensive application in RL (Beck et al., 2025). Historically, RL training pipelines have been constrained by CPU-bound simulators. The advent of GPU-accelerated environments, however, has enabled speedups of up to $1000\times$, revitalizing interest in Meta-RL. This has facilitated the discovery of novel loss functions (Lu et al., 2022; Jackson et al., 2024), activation functions (Nadimpalli et al., 2025), and optimizers (Goldie et al., 2024) for Deep RL. Nevertheless, differentiating through the learning process of an RL algorithm remains a challenging problem, motivating the use of black-box optimization methods (Salimans et al., 2017; Chen et al., 2023b). However, these approaches typically suffer from high sample complexity and limited interpretability. Large language models (LLMs) provide a promising alternative as their broad domain knowledge and code-generation capabilities can enable more interpretable and effective learning algorithms (Goldie et al., 2025). Beyond RL, LLM-based black box optimization has demonstrated success in a wide range of domains, including environment generation (Faldor et al., 2024), neural architecture search (Chen et al., 2023a), combinatorial optimization (Ye et al., 2024), mathematics (Romera-Paredes et al., 2024), and more (Novikov et al., 2025).

## 3  BACKGROUND

**Preliminaries** We consider a Markov Decision Process (MDP) $\mathcal{M}$ defined by the tuple $(\mathcal{S}, \mathcal{A}, \mathcal{P}, r, \gamma, \mu)$, where $\mathcal{S}$ denotes the set of states, $\mathcal{A}$ is the set of actions, $\mathcal{P}(s'|s,a) \in [0,1]$ is the transition probability, $r(s,a) \in \mathbb{R}$ is the reward function, $\gamma \in [0,1]$ is the discount factor, and $\mu(s) \in \Delta(\mathcal{S})$ is the initial state distribution. A policy $\pi(.|s) \in \Delta(\mathcal{A})$ is a distribution over the set of valid actions for state $s$. A *trajectory* $\tau = \{(s_t, a_t)\}$ denotes the state-action pairs encountered by executing $\pi$ in $\mathcal{M}$. For a given policy $\pi$, the occupancy measure $\rho_\pi(s,a)$ is defined as $\rho^\pi(s,a) = (1-\gamma)\mathbb{E}_{\tau \sim \pi}[\sum_{t=0}^{\infty} \gamma^t P(s_t = s, a_t = a)]$. Intuitively, it can be interpreted as the distribution over state-action pairs that the agent encounters while following policy $\pi$. A one-to-one correspondence exists between $\pi$ and $\rho_\pi$ (Syed et al., 2008), allowing us to use them interchangeably. For functions $f(s,a)$ dependent only on state-action pairs, we have $\mathbb{E}_{\tau \sim \pi}\left[\sum_{t=0}^{\infty} \gamma^t f(s_t, a_t)\right] = \frac{1}{1-\gamma}\mathbb{E}_{(s,a)\sim \rho_\pi}[f(s,a)]$. We leverage this identity to use the two expectations interchangeably when optimizing over $f$.

**Distribution Matching IL**   The goal of *distribution-matching imitation learning* is to find a policy $\pi^*$ whose occupancy measure $\rho_\pi$ closely aligns with that of the expert $\rho_E$. In this paper, we focus on $f$-divergence minimization—a unifying framework encompassing many distribution-matching IL algorithms (Ghasemipour et al., 2020). Formally,

$$\pi^* = \arg\min_\pi \ D_f(\rho_E \| \rho_\pi), \tag{1}$$

where the $f$-divergence $D_f$ is defined as

$$D_f(\rho_E \| \rho_\pi) = \mathbb{E}_{(s,a)\sim\rho_\pi} \left[ f \left( \frac{\rho_E(s,a)}{\rho_\pi(s,a)} \right) \right], \tag{2}$$

with $f : \mathbb{R}_+ \to \mathbb{R}$ convex and satisfying $f(1) = 0$. The ratio $\frac{\rho_E}{\rho_\pi}$, known as the *density ratio*, is formally the *Radon–Nikodym derivative* (Halmos, 1950), which quantifies the pointwise discrepancy between $\rho_E$ and $\rho_\pi$. In practice, we only have access to a finite set of expert demonstrations rather than the true occupancy measure $\rho_E$.

**Adversarial Imitation Learning**   An adversarial approach to solving Equation 1 involves the following iterative steps: **(1) Policy Rollouts**: Generate trajectories by executing the current policy $\pi$. **(2) Density Ratio Estimation**: Multiple approaches have been proposed to estimate the density ratio $\frac{\rho_E}{\rho_\pi}$ (Wang et al., 2024; Lai et al., 2024). In this work, we adopt the most common approach (Orsini et al., 2021) of training a classifier (discriminator) to distinguish expert from policy-generated $(s,a)$ pairs via binary cross-entropy loss. An optimal discriminator's logits (pre-softmax) would then correspond to $\log \left( \frac{\rho_E}{\rho_\pi} \right)$ (refer to Appendix A.1 for derivation) and **(3) Reward Assignment and Policy Improvement**: Depending on the $f$-divergence being minimized, each $(s,a)$ pair visited by $\pi$ receives a reward $r(s,a) = r_f \left( \frac{\rho_E(s,a)}{\rho_\pi(s,a)} \right)$, where $r_f : \mathbb{R}_+ \to \mathbb{R}$ is defined as the reward assignment function. Table 1 summarizes common divergences and their corresponding reward assignment functions. The rewards are then used to update the policy, and **Steps 1-3 are repeated until convergence**.

Table 1: Reward assignment functions for different $f$-divergences, where $\ell = \log \frac{\rho_E(s,a)}{\rho_\pi(s,a)}$

| Divergence | Algorithm | Reward Assignment Function |
|---|---|---|
| Forward KL | FAIRL (Ghasemipour et al., 2020) | $-\ell(s,a) \cdot e^{\ell(s,a)}$ |
| Backward KL | AIRL (Fu et al., 2018) | $\ell(s,a)$ |
| Jensen-Shannon | GAIL (Ho & Ermon, 2016) | $\text{softplus}(\ell(s,a))$ |
| Unnamed $f$-div | GAIL-heuristic (Orsini et al., 2021) | $-\text{softplus}(-\ell(s,a))$ |

## 4   PROBLEM DEFINITION

Adversarial methods are often considered unstable due to their reliance on optimizing min-max objectives, akin to GANs (Goodfellow et al., 2014). To mitigate this instability, prior work has predominantly focused on **Step 2** by improving discriminator training (Luo et al., 2024; Wang et al., 2024).

In this paper, we focus on **Step 3**, highlighting that providing an informative learning signal is crucial for effective policy improvement and overall adversarial training. Originally discussed by Ghasemipour et al. (2020), Figure 1 illustrates how the reward assignment function shapes the policy's learning dynamics. The AIRL RA function encourages the policy to visit state-action pairs where expert visitation exceeds its own and penalizes it equally when it surpasses the expert. In contrast, the GAIL RA function *only* incentivizes matching the expert on underrepresented pairs, while its heuristic variant does the opposite. FAIRL employs a more nuanced approach: it rewards the policy for slightly exceeding expert visitation, but imposes steep penalties when the expert dominates. Ghasemipour et al. (2020) argues that this drives the policy to gradually expand and cover

---

While the original convex function $f$ can be applied directly, most approaches utilize the function derived from its variational representation, as detailed in the Appendix A

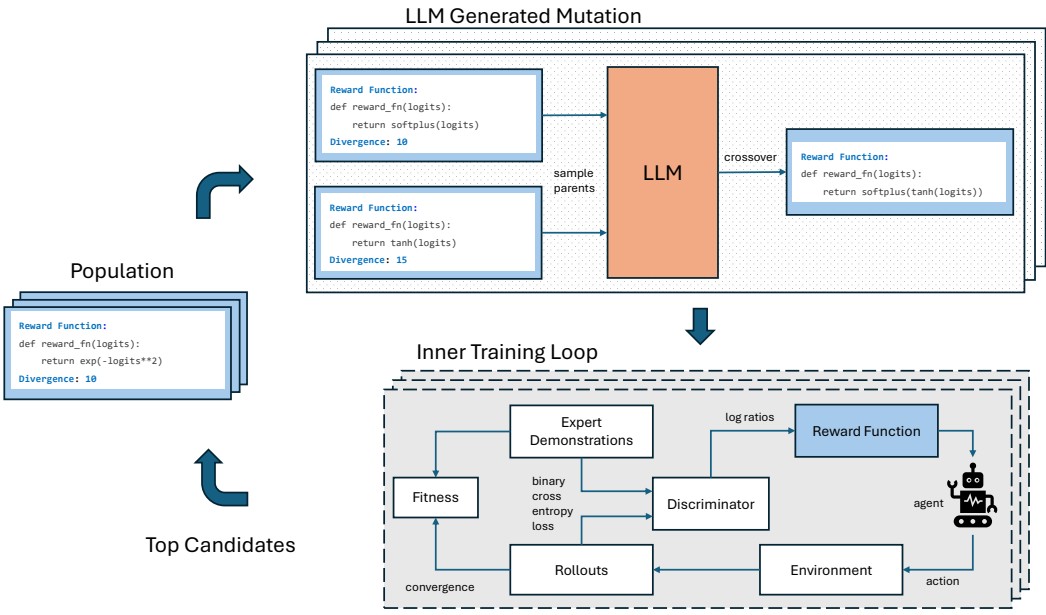

Figure 2: Visualization of the LLM-guided evolution. Appendix B contains the pseudocode of the framework.

the expert distribution from the outside in. Furthermore, although existing RA functions (Table 1) are derived from well-established $f$-divergence theory, they neglect the practical stability challenges that arise during training.

This raises the question: *Can we meta-learn a reward assignment function that results in stable and effective adversarial training?* To evaluate training quality, we measure the divergence between the expert and the policy after training. Specifically, we use the Wasserstein distance (Rubner et al., 1998), computed between rollouts generated by the policy and expert demonstrations. We chose this metric due to its robustness and sensitivity in measuring distances between occupancy distributions in RL settings (Luo et al., 2023; Rupf et al., 2024).

### 4.1 FORMAL DEFINITION

We formalize the meta-learning problem of discovering RA functions as:

$$\min_{f} \ \mathcal{W}(\rho_E, \rho_{\pi^*}; f) \quad \text{s.t.} \quad \pi^* = \arg\max_{\pi} r_f(\rho_E \| \rho_\pi), \tag{3}$$

where $\mathcal{W}$ denotes the Wasserstein distance, and $\pi^*$ is the optimal policy obtained by iterating over Steps 1–3 with reward assignment function $r_f$. We remove additional constraints on $r_f$ (such as convexity), to enable the exploration of more expressive RA functions beyond those derivable from classical $f$-divergences. While this approach foregoes theoretical convergence guarantees, the trade-off is justified by the empirical feedback that $r_f$ receives with policy training. As our results show, the discovered reward assignment functions exhibit robust generalization properties.

## 5 DISCOVERING RA FUNCTIONS VIA EVOLUTIONARY SEARCH

Optimizing Equation 3 is challenging because it requires backpropagating gradients through the entire adversarial training loop, which is generally computationally intractable. Consequently, prior work has typically relied on black-box methods for such bilevel optimization problems (Goldie et al., 2024; Lu et al., 2022). In this work, we adopt an **LLM-guided evolutionary** framework that has been shown to be sample-efficient, interpretable, and capable of discovering generalizable algorithms in meta-RL (Goldie et al., 2025).

The RA function $r_f$ is represented directly as code, enabling expressive and interpretable formulations. To evolve new candidate functions, we prompt the LLM to intelligently combine and mutate parent programs, guided by their structural and behavioral characteristics. LLMs are particularly well-suited for this setting for two key reasons: (1) code (Python) is Turing-complete (Faldor et al., 2024), allowing the search space to encompass a rich class of reward assignment functions and (2) the pretrained knowledge encoded in LLMs provides a strong inductive bias, helping to navigate the vast search space more effectively.

Next, we describe the LLM-guided search algorithm used in our work. Importantly, our main contribution lies not in the evolutionary algorithm itself, but in the formulation and optimization of the meta-learning objective (Eq. 3). LLM-guided black-box optimization does not follow a rigid standard—many approaches share a common structure but differ in finer details. For completeness, we outline the variant adopted in this work, which closely resembles EvolveAct (Nadimpalli et al., 2025), Evo-prompting (Chen et al., 2023a), and FunSearch (Romera-Paredes et al., 2024). More sophisticated strategies (Fernando et al., 2024; Novikov et al., 2025) are complementary to this work.

## 5.1 COMPONENTS OF THE EVOLUTIONARY SEARCH

**Base Population.** We initialize the search with reward assignment functions from established $f$-divergences—GAIL, FAIRL, AIRL, and GAIL-heuristics (Table 1)—to provide a robust foundation for the evolutionary search.

**Fitness Evaluation.** Each candidate function $r_f$ is evaluated by training a policy to convergence using $r_f$ as the reward assignment function, then measuring the Wasserstein distance between its rollouts and the expert's. This score serves as the fitness criterion for selection.

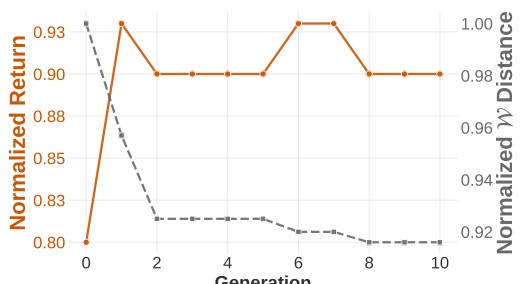

Figure 3: Performance across generations on the *Minatar SpaceInvaders* environment. We report the best-performing member per generation, with Generation $0$ denoting the base population. $\mathcal{W}$ distance is normalized relative to the best base member (GAIL).

**Crossover.** To generate new candidate functions, we sample parent pairs $\{r_{f_1}, r_{f_2}\}$ from the current population and pass them to the LLM along with their fitness scores. The LLM is then prompted to synthesize a new function $r_{f_3}$ that combines desirable properties of the parents, with the goal of improving performance (e.g., by blending their functional forms). The detailed prompt format and representative examples of generated candidates are provided in Appendix C.

## 5.2 SEARCH PROCEDURE

The LLM-guided evolutionary framework unfolds over multiple iterations as follows:

**Initial Population:** Initialize the search with a base population of reward assignment functions derived from known $f$-divergence formulations.

**Iterative Evolution:** At each generation: (1) Randomly sample $M$ pairs of reward assignment functions from the current population. (2) For each pair, use the LLM to generate $N$ new candidate functions by recombining and refining the parent functions. (3) Evaluate all $M \times N$ candidates using the distribution-matching fitness score. (4) Select the top $K$ candidates to populate the next generation.

**Termination:** Repeat until a stopping criterion is achieved (e.g., performance plateau).

## 6 EMPIRICAL STUDIES

We begin by describing the experimental setup, after which we present the results. We conducted our experiments on two benchmark suites: MuJoCo control tasks (*Ant, Reacher, Walker2d, HalfChee-*

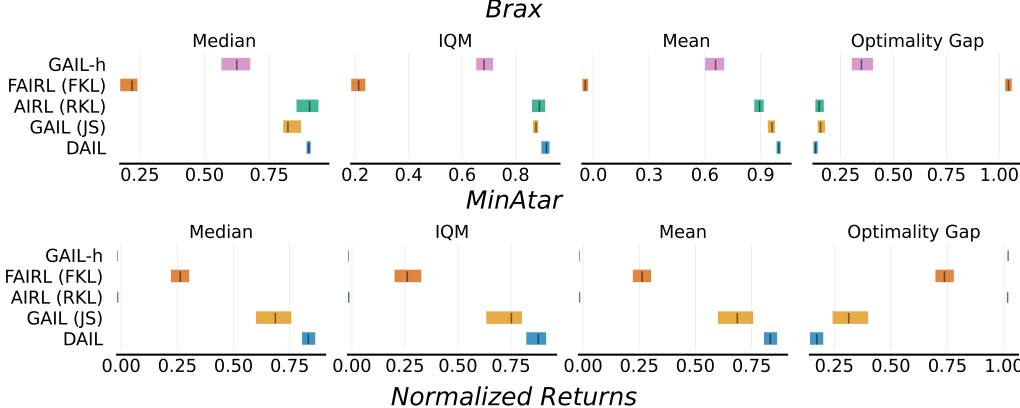

Figure 4: Aggregate performance on the Brax and Minatar suites (excluding SpaceInvaders).

*tah, and Hopper*) (Todorov et al., 2012) and Minatar (*Asterix, SpaceInvaders, and Breakout*) (Young & Tian, 2019). For each task, we collect 10 *successful* expert demonstrations from a PPO-trained policy (Schulman et al., 2017) and subsample every $20^{th}$ transition, following standard practice (Ho & Ermon, 2016). Unless stated otherwise, we optimize the policy using PPO and regularize the discriminator with a gradient penalty (Gulrajani et al., 2017). Consistent with prior work in IL, we adopt fixed-length episodes of 1000 timesteps for MuJoCo environments (Gleave et al., 2022). However, we do not enforce this for the Minatar tasks since doing so significantly degraded performance. Our implementation is fully written in JAX (Bradbury et al., 2018), using PureJaxRL (Lu et al., 2022), Brax (Freeman et al., 2021) for MuJoCo environments, Gymnax (Lange, 2022) for Minatar, and OTT-JAX (Cuturi et al., 2022) for computing Wasserstein distance. The complete training hyperparameters are provided in Appendix D. We normalize returns using min-max scaling between random and expert policy performance, with all results averaged over 16 independent seeds.

### 6.1 EVOLUTIONARY SEARCH

We performed the evolutionary search on the *Minatar SpaceInvaders* environment, which has previously been shown to facilitate the discovery of generalizable meta-RL algorithms (Jackson et al., 2024). For the search, we use GPT-4.1-mini, selected for its strong performance–cost tradeoff. Throughout evolution, we fix the PPO and discriminator hyperparameters, evaluating 200 candidate RA functions over a span of three hours. Full details of the evolutionary hyperparameters are provided in Appendix D.

The evolutionary trajectory is shown in Figure 3. The best-performing reward assignment function discovered at the end of the search is:

$$r_{\text{disc}}(x) = 0.5 \cdot \text{sigmoid}(x) \cdot [\tanh(x) + 1]. \tag{4}$$

Building on this, we introduce *Discovered Adversarial Imitation Learning* (DAIL), which applies a standard imitation learning loop with $r_{\text{disc}}$ as the reward assignment function. Remarkably, DAIL reduces the $\mathcal{W}$-distance to expert trajectories by **20%** and improves normalized returns by **12.5%** compared to the best baseline (GAIL).

### 6.2 GENERALIZATION

We now turn to the out-of-distribution performance of DAIL. As a first step, we benchmark our method against previous RA functions (Table 1) in Brax and Minatar environments (excluding Minatar SpaceInvaders). All methods share identical hyperparameters, differing only in the choice of RA function. Aggregated scores using the `rliable` library (Agarwal et al., 2021) are shown in Figure 4. Among the baselines, AIRL performs slightly better than GAIL on Brax but performs

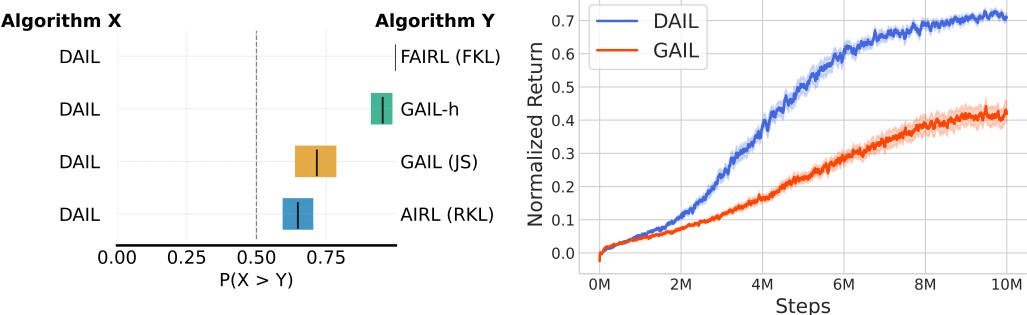

Figure 5: (Left) Probability of improvement of DAIL over baselines on Brax. (Right) Performance comparison between DAIL and GAIL (on Minatar SpaceInvaders) using A2C. We report the mean and standard error (SEM).

poorly on Minatar, similar to GAIL-heuristic. This is likely because their reward assignment functions yield predominantly negative rewards, incentivizing agents to terminate episodes early. FAIRL performs poorly across both suites, likely due to its exponential, unbounded reward decay for positive log-ratios, which destabilizes training. Overall, the baseline trends align with the observations from (Orsini et al., 2021).

Figure 4 demonstrates DAIL's effectiveness across both benchmark suites. On Minatar, DAIL significantly outperforms all baselines across all evaluation metrics. On Brax, DAIL outperforms baselines on most metrics, with a slightly lower median than AIRL and statistically significant gains in mean performance. To quantify the robustness of these improvements on Brax, we employ the *probability of improvement* (Agarwal et al., 2021) metric, which estimates the probability that DAIL outperforms a baseline on a randomly chosen task. As shown in Figure 5 (left), DAIL achieves a probability of improvement greater than 0.5 against all baselines, providing further evidence of its superior performance. To assess DAIL's generalization beyond the policy optimizer (PPO) used during evolution, we evaluate its performance with A2C (Mnih et al., 2016). As shown in Figure 5 (right), DAIL maintains significant performance advantages over GAIL showcasing its effectiveness across different policy optimization algorithms. Finally, we assess DAIL's generalization performance under various discriminator regularization strategies proposed by Orsini et al. (2021), including weight decay, an additional entropy bonus, and spectral normalization of the discriminator weights. We also consider the case without any regularization. As shown in Table 2, DAIL outperforms GAIL in 3 out of 5 regularization regimes.

| Algo | Env | none | w-decay | entropy | spectral | grad-pen |
|------|-----|------|---------|---------|----------|----------|
| DAIL | Asterix | $0.88 \pm 0.03$ | $1.33 \pm 0.03$ | $0.12 \pm 0.01$ | $0.92 \pm 0.03$ | $0.66 \pm 0.03$ |
| | Breakout | $0.81 \pm 0.07$ | $0.74 \pm 0.08$ | $0.91 \pm 0.02$ | $0.77 \pm 0.07$ | $1.01 \pm 0.00$ |
| | SpaceInvaders | $0.71 \pm 0.07$ | $0.81 \pm 0.01$ | $0.80 \pm 0.01$ | $0.70 \pm 0.09$ | $0.90 \pm 0.00$ |
| | **Overall** | $0.80 \pm 0.03$ | $\mathbf{0.96} \pm \mathbf{0.03}$ | $0.61 \pm 0.01$ | $\mathbf{0.80} \pm \mathbf{0.04}$ | $\mathbf{0.85} \pm \mathbf{0.01}$ |
| GAIL | Asterix | $1.18 \pm 0.03$ | $1.44 \pm 0.03$ | $0.48 \pm 0.03$ | $0.22 \pm 0.03$ | $0.52 \pm 0.04$ |
| | Breakout | $0.76 \pm 0.07$ | $0.52 \pm 0.10$ | $0.89 \pm 0.01$ | $0.33 \pm 0.10$ | $0.85 \pm 0.07$ |
| | SpaceInvaders | $0.61 \pm 0.09$ | $0.34 \pm 0.09$ | $0.81 \pm 0.00$ | $0.42 \pm 0.08$ | $0.81 \pm 0.03$ |
| | **Overall** | $\mathbf{0.85} \pm \mathbf{0.04}$ | $0.76 \pm 0.04$ | $\mathbf{0.73} \pm \mathbf{0.01}$ | $0.32 \pm 0.05$ | $0.73 \pm 0.03$ |

Table 2: Performance of DAIL and GAIL under different discriminator regularization strategies. Hyperparameters are adopted from Orsini et al. (2021). Reported values denote the mean and standard error across runs.

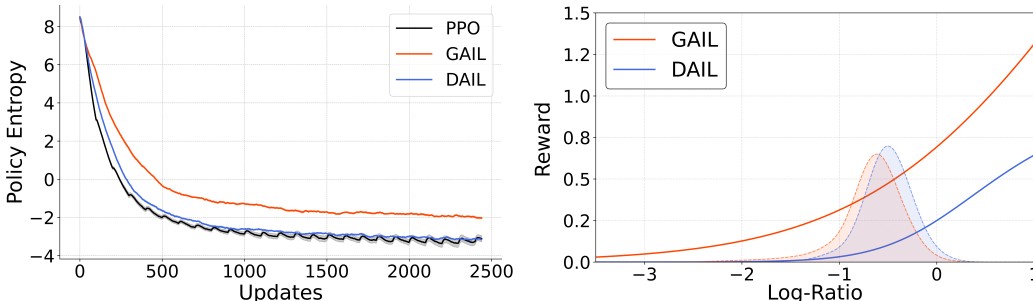

Figure 6: (Left) Policy entropy during training (on HalfCheetah) with different RA functions; PPO with simulator rewards is presented as reference. Results show mean ± SEM. (Right) Distribution of log-density ratios during training on Minatar SpaceInvaders, estimated via kernel density fitting.

## 6.3 ANALYSIS

Next, we investigate why DAIL leads to such strong performance. As shown in Figure 1, $r_{\text{disc}}$ exhibits an S-shape with a sharper gradient and a slight rightward shift compared to a standard sigmoid. Importantly, it is bounded within the interval $[0, 1]$, unlike existing baselines. Prior work has shown that bounding rewards can stabilize Deep RL (Mnih et al., 2015; Van Hasselt et al., 2016), which we hypothesize contributes to DAIL's performance.

We assess training stability by tracking policy entropy and comparing it to a PPO agent with access to simulator rewards. As shown in Figure 6 (left), policies trained with $r_{\text{disc}}$ converge to lower entropy, closely matching the PPO baseline. This suggests that $r_{\text{disc}}$ delivers a rich and informative signal, enabling sharper action distributions that reflect confident behavior and effective reward maximization. In contrast, GAIL's RA function produces noisier signals, leading to higher-entropy policies and greater uncertainty in action selection.

To gain deeper insight, we examine the distribution of log density ratios $\log \frac{\rho_E}{\rho_\pi}$ of state-action pairs visited during training. We compare the distributions between DAIL and GAIL and analyze the interaction with their respective RA functions. Figure 6 (right) shows that a large fraction of the log-ratios lie within the interval $[-1, 0]$ for both methods, with a long tail extending to approximately $-2$—a region we identify as indicative of random policy behavior. $r_{\text{disc}}$ saturates near zero for $x \lesssim -1.8$, effectively filtering out noisy or low-quality state-action pairs while maintaining informative gradients for moderately performing ones. In contrast, GAIL's reward function assigns high positive values even at $x = -2$, thereby rewarding state-actions pairs corresponding to near-random policies. We posit that this over-sensitivity to low-quality behavior contributes to the noisier reward signals and instability observed in GAIL's training dynamics. Note that the findings in Figure 6 generalize across all test environments; results for which are omitted due to space constraints.

To further test this hypothesis, we conduct an ablation study on the individual components of $r_{disc}$, comparing it against $\text{sigmoid}(x)$ and $0.5 \cdot [\tanh(x) + 1]$. All three functions map to [0,1] and exhibit S-shaped curves, but differ in their response characteristics: $r_{disc}$ closely follows the tail of the density ratio distribution, while the other two functions provide noisier reward signals that remain positive around $x = -1.8$, with $\text{sigmoid}(x)$ being the least responsive due to its relatively flat profile. These differences are validated empirically (Figure 7), where $r_{disc}$ achieves the best performance, followed by $0.5 \cdot [\tanh(x) + 1]$, with $\text{sigmoid}(x)$ performing worst.

## 6.4 STABILITY

To assess the stability of the evolutionary process, we conduct an additional independent evolutionary run on the *Minatar SpaceInvaders* environment. The top-5 RA functions discovered in both runs are presented in Table 8 and the top-3 are plotted in Figure 8. We see that the 6 evolved RA functions exhibit highly similar structures. Further, they maintain informative gradients within the range $[-1, 0]$ and saturate rapidly thereafter, consistent with our analysis in Section 6.3. Further-

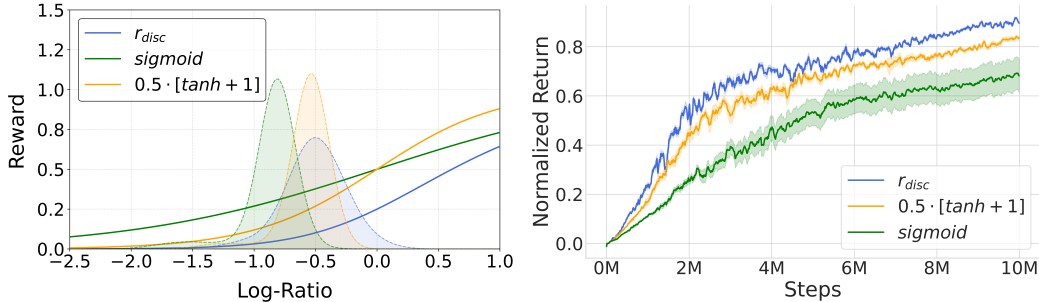

Figure 7: (Left) Log-density ratio distributions during training (Minatar SpaceInvaders), estimated via kernel density estimation. (Right) $r_{disc}$ vs. component function performance (Minatar SpaceInvaders).

more, we conduct an evolutionary run in the *MinAtar Breakout* environment and find that DAIL emerges among the top-5 RA functions in the final generation, demonstrating both the stability of the evolutionary process and the effectiveness of DAIL.

# 7 CONCLUSION

**Summary**  This work highlights the importance of the RA functions in influencing both policy optimization and the overall stability of AIL—an aspect that has received relatively little attention. We introduce a novel approach using LLM-guided evolutionary search to automatically discover optimal reward assignment functions, resulting in DAIL, the first meta-learned AIL algorithm. Experimental validation demonstrates DAIL's superior and consistent performance across unseen environments and policy optimization algorithms. Through analysis of DAIL's discovered reward function $r_{disc}$ and its impact on training dynamics, we provide novel insights into these performance improvements.

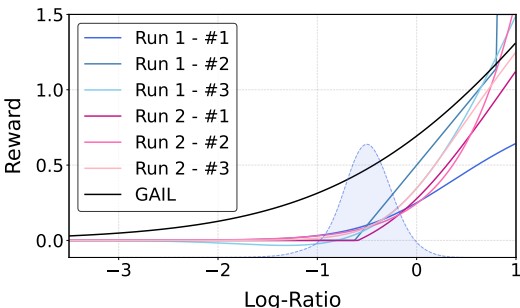

Figure 8: Comparing the top-3 discovered RA functions across two independant evolutionary runs. Note that *Run 1 - #1* corresponds to DAIL.

**Limitations and Future Work**  While DAIL demonstrates strong generalization, the discovered RA function $r_{\text{disc}}$ does not correspond to a valid $f$-divergence and therefore lacks theoretical guarantees. Moreover, despite its strong performance, the RA function of DAIL remains *static* throughout training and does not adapt to the training state (e.g., number of updates remaining, loss, observed log-ratios). Exploring *time-aware* RA functions—those that condition on the training state—could yield richer, more informative learning signals, similar to Jackson et al. (2024). Additionally, including more information into the LLM's context such as environment information may facilitate more effective crossovers.

Finally, it would be valuable to evaluate DAIL's generalization to more complex benchmarks, such as the Atari-57 (Bellemare et al., 2013) and Procgen (Cobbe et al., 2020) suites. Prior work in meta-learning Oh et al. (2020) has demonstrated that training across a diverse set of environments yields more robust algorithms. For instance, Oh et al. (2025) introduced the Disco57 and Disco103 suites, consisting of 57 and 103 environments, respectively. Investigating the discovery of AIL algorithms meta-trained on such large-scale environment collections is an exciting direction for future research, though it currently lies beyond our computational budget.

## 8 ETHICS STATEMENT

As with other meta-learning approaches, the automated discovery of the algorithms (DAIL) obscures its properties and behavior, making analyses like those in Section 6.3 both challenging and essential for understanding such algorithms. Additionally, as with other IL algorithms, these advances hold promise for safer and more capable AI systems but also introduce risks, including misuse (e.g., imitation of harmful behaviors) and bias (inherited from expert data).

## 9 REPRODUCIBILITY STATEMENT

All hyperparameters and experimental details are reported in Section 6 and Appendix D, with complete information on prompts and the LLM in Appendix C. The supplementary material provides code (including plotting scripts) and evaluation data to fully reproduce all the main results in this work.

## 10 ACKNOWLEDGMENTS

**This research/project was supported by the Singapore Ministry of Education (MOE) Academic Research Fund (AcRF) Tier 1 grant (Proposal ID: 24-SIS-SMU-010).**

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

# A ON $f$-DIVERGENCE MINIMIZATION

We present key preliminary results that will support the derivations in later sections.

## A.1 BACKGROUND

Note that the results presented here assume a discounted infinite-horizon setting in a discrete MDP for simplicity, but they can be extended to other settings such as finite-horizon problems or continuous state-action spaces.

**Definition** (Occupancy Measure) For a policy $\pi$, the occupancy measure $\rho_\pi$ is defined as:

$$\rho^\pi(s,a) = (1-\gamma)\mathbb{E}_{\tau\sim\pi}\left[\sum_{t=0}^{\infty}\gamma^t\mathbb{P}(s_t = s, a_t = a)\right] \tag{1}$$

**Lemma 1** (Interchange of Expectations). *For any scalar function $f : S \times A \to \mathbb{R}$ and discount factor $\gamma \in [0,1)$,*

$$\mathbb{E}_{\tau\sim\pi}\left[\sum_{t=0}^{\infty}\gamma^t f(s_t, a_t)\right] = \frac{1}{1-\gamma}\mathbb{E}_{(s,a)\sim\rho_\pi}\left[f(s,a)\right]. \tag{2}$$

*Proof.* Starting from the definition of the left-hand side, we have

$$\mathbb{E}_{\tau\sim\pi}\left[\sum_{t=0}^{\infty}\gamma^t f(s_t, a_t)\right] = \mathbb{E}_{\tau\sim\pi}\left[\sum_{t=0}^{\infty}\gamma^t\left(\sum_{s,a}\mathbb{P}(s_t = s, a_t = a)f(s,a)\right)\right] \tag{3}$$

$$= \sum_{s,a}\left[\left(\mathbb{E}_{\tau\sim\pi}\sum_{t=0}^{\infty}\gamma^t\mathbb{P}(s_t = s, a_t = a)\right)f(s,a)\right] \tag{4}$$

$$= \sum_{s,a}\frac{\rho_\pi(s,a)}{1-\gamma}f(s,a) \quad \text{(using Eq. 1)} \tag{5}$$

$$= \frac{1}{1-\gamma}\mathbb{E}_{(s,a)\sim\rho_\pi}\left[f(s,a)\right] \tag{6}$$

$\square$

**Lemma 2** (Optimal Discriminator). *Let $P$ and $Q$ be two distributions over a random variable $X$. Consider a discriminator parameterized as $D(X) = \sigma(\ell(X))$, where $\sigma$ denotes the sigmoid function and $D(X)$ represents the predicted probability that $X$ is drawn from $P$. If the discriminator is trained via likelihood maximization (i.e., using binary cross-entropy loss), then the optimal discriminator satisfies:*

$$\ell^*(X) = \log\left(\frac{P(X)}{Q(X)}\right). \tag{7}$$

*Proof.* The discriminator is trained using the binary cross-entropy loss:

$$\mathcal{L}_{\text{BCE}} = \mathbb{E}_{X\sim P}[\log D(X)] + \mathbb{E}_{X\sim Q}[\log(1 - D(X))]. \tag{8}$$

To find the optimal discriminator, we maximize $\mathcal{L}_{\text{BCE}}$ with respect to $D(X)$ pointwise. Taking the derivative of the pointwise objective and setting it to zero yields:

$$D^*(X) = \frac{P(X)}{P(X) + Q(X)}. \tag{9}$$

Substituting $D^*(X) = \sigma(\ell^*(X))$, we get:

$$\sigma(\ell^*(X)) = \frac{1}{1 + e^{-\ell^*(X)}} = \frac{P(X)}{P(X) + Q(X)}. \tag{10}$$

Solving for $\ell^*(X)$ gives:

$$\ell^*(X) = \log\left(\frac{P(X)}{Q(X)}\right). \tag{11}$$

$\square$

## A.2 FAIRL REWARD ASSIGNMENT FUNCTION

The $f$-divergence between two distributions $P$ and $Q$ is defined as:

$$D_f(P\|Q) = \mathbb{E}_{X \sim Q}\left[f\left(\frac{P(X)}{Q(X)}\right)\right],\tag{12}$$

where $f : \mathbb{R}_+ \to \mathbb{R}$ is a convex function satisfying $f(1) = 0$. The ratio $\frac{P(X)}{Q(X)}$ is referred to as the *density ratio*, and is formally defined as the *Radon-Nikodym derivative* (Halmos, 1950) of $P$ with respect to $Q$.

In the context of $f$-divergence-based imitation learning, the objective is to find a policy $\pi$ that minimizes the divergence between the expert and policy occupancy measures:

$$\pi^* = \arg\min_{\pi}\ D_f(\rho_E\|\rho_\pi)\tag{13}$$

$$= \arg\min_{\pi}\ \mathbb{E}_{(s,a)\sim\rho_\pi}\left[f\left(\frac{\rho_E(s,a)}{\rho_\pi(s,a)}\right)\right]\tag{14}$$

$$= \arg\min_{\pi}\ (1-\gamma)\cdot\mathbb{E}_{\tau\sim\pi}\left[f\left(\frac{\rho_E(s,a)}{\rho_\pi(s,a)}\right)\right]\quad\text{(using Lemma 1)}\tag{15}$$

$$= (1-\gamma)\cdot\arg\max_{\pi}\ \mathbb{E}_{\tau\sim\pi}\left[r_f(s,a)\right],\tag{16}$$

where the reward function $r_f$ is defined as:

$$\boxed{r_f(s,a) = -f\left(\frac{\rho_E(s,a)}{\rho_\pi(s,a)}\right)}\tag{17}$$

This formulation establishes that minimizing an $f$-divergence is equivalent to maximizing the expected return under a reward assignment function $r_f$ which is a mapping from the *density ratio* $\frac{\rho_E}{\rho_\pi}$ to a scalar reward.

In the case of reverse KL divergence, the corresponding $f$-function is $f(x) = x\log x$. Defining the log-density ratio as $\ell(s,a) = \log\left(\frac{\rho_E(s,a)}{\rho_\pi(s,a)}\right)$, the resulting reward assignment becomes:

$$\boxed{r_{f\text{-RKL}}(s,a) = -e^{\ell(s,a)}\cdot\ell(s,a)}\tag{18}$$

which corresponds to the reward used in FAIRL (Ghasemipour et al., 2020).

## A.3 REWARD ASSIGNMENT FUNCTIONS IN GAIL AND AIRL

The $f$-divergence admits a variational representation:

$$D_f(P\|Q) = \sup_{g:\mathcal{X}\to\mathbb{R}} \mathbb{E}_{X\sim P}[g(X)] - \mathbb{E}_{X\sim Q}[f^*(g(X))],\tag{19}$$

where $f^*$ denotes the convex conjugate of $f$, defined by

$$f^*(u) = \sup_{v\in\text{dom}(f)}\{uv - f(v)\}.$$

To understand the structure of the optimal function $g$, we consider the pointwise optimization of the integrand in Eq. equation 19. Letting $u = g(X)$ and $c = \frac{P(X)}{Q(X)}$, the first-order optimality condition becomes:

$$\nabla_u f^*(u) = c.\tag{20}$$

Under the assumption that $f$ is differentiable and strictly convex, the gradient of the convex conjugate satisfies the inverse relationship:

$$\nabla f^*(u) = (\nabla f)^{-1}(u).\tag{21}$$

Substituting into the optimality condition, we obtain:

$$(f')^{-1}(u) = c \tag{22}$$

$$\Rightarrow \quad u = f'(c). \tag{23}$$

Thus, we have,

$$\boxed{g^*(X) = f'\left(\frac{P(X)}{Q(X)}\right)} \tag{24}$$

In the context of $f$-divergence based imitation learning, we have,

$$\pi^* = \arg\min_{\pi} \; D_f(\rho_\pi \| \rho_E) \tag{25}$$

$$= \arg\min_{\pi} \; \sup_{g} \mathbb{E}_{(s,a)\sim\rho_\pi}[g(s,a)] - \mathbb{E}_{(s,a)\sim\rho_{\pi_e}}[f^*(g(s,a))] \tag{26}$$

$$= \arg\min_{\pi} \; \mathbb{E}_{(s,a)\sim\rho_\pi}\left[f'\left(\frac{\rho_\pi(s,a)}{\rho_E(s,a)}\right)\right] \quad \text{(using Eq. 24)} \tag{27}$$

$$= (1-\gamma) \cdot \arg\max_{\pi} \; E_{\tau\sim\pi}[r_{f\text{-var}}(s,a)] \tag{28}$$

where the reward function $r_{f\text{-var}}$ is defined as:

$$\boxed{r_{f\text{-var}}(s,a) = -f'\left(\frac{\rho_E(s,a)^{-1}}{\rho_\pi(s,a)}\right)} \tag{29}$$

This establishes that minimizing the $f$-divergence between $\rho_\pi$ and $\rho_E$ via its variational formulation is equivalent to maximizing the expected return under a reward assignment function defined by $r_{f\text{-var}}$.

In the case of reverse-KL divergence, the corresponding $f$-function is $f(x) = x \cdot \log x$, and thus $f'(x) = 1 + \log x$. Plugging this into the reward assignment gives:

$$r_{f\text{-var-RKL}}(s,a) = -\left(1 + \log\left(\left(\frac{\rho_E(s,a)}{\rho_\pi(s,a)}\right)^{-1}\right)\right) \tag{30}$$

$$= \log\left(\frac{\rho_E(s,a)}{\rho_\pi(s,a)}\right) - 1 \tag{31}$$

Ignoring the additive constant, the reward assignment under the reverse-KL divergence simplifies to:

$$\boxed{r_{f\text{-var-RKL}}(s,a) = \ell(s,a)} \tag{32}$$

which corresponds to the reward used in AIRL (Fu et al., 2018).

In the case of Jensen-Shannon divergence, the corresponding $f$-function is $f(x) = -(x+1)\log\left(\frac{x+1}{2}\right) + x\log x$, and its derivative $f'(x) = \log\left(\frac{2x}{x+1}\right)$. Hence the variational reward assignment becomes:

$$r_{f\text{-var-JS}}(s,a) = -\log\left(\frac{2\left(\rho_\pi(s,a)/\rho_E(s,a)\right)}{1 + \rho_\pi(s,a)/\rho_E(s,a)}\right) \tag{33}$$

$$= \log\frac{1}{2}\left(1 + \frac{\rho_E(s,a)}{\rho_\pi(s,a)}\right) \tag{34}$$

Ignoring the additive constant, the reward assignment under the JS divergence simplifies to:

$$\boxed{r_{f\text{-var-JS}}(s,a) = \log(1 + e^{\ell(s,a)})} \tag{35}$$

which corresponds to the reward used in GAIL (Ho & Ermon, 2016).

By plugging the $f$-functions of other commonly used $f$-divergences into Eq. 29, we can derive their corresponding reward assignment functions, as summarized in Table 3. While no formal adversarial imitation learning (AIL) methods explicitly employ these divergences, they have been explored in the context of non-adversarial imitation learning algorithms such as IQ-Learn (Garg et al., 2021).

Table 3: Reward assignment functions derived from different $f$-divergences

| Divergence | $f(x)$ | $r_{f\text{-var}}$ |
|---|---|---|
| Forward KL | $-\log x$ | $\frac{\rho_E}{\rho_\pi}$ |
| Reverse KL | $x \log x$ | $\log \frac{\rho_E}{\rho_\pi} - 1$ |
| Jensen-Shannon | $x \log x - (x+1)\log\left(\frac{x+1}{2}\right)$ | $\log \frac{1}{2}\left(1 + \frac{\rho_E}{\rho_\pi}\right)$ |
| Squared Hellinger | $(\sqrt{x}-1)^2$ | $\sqrt{\frac{\rho_E}{\rho_\pi}} - 1$ |
| Pearson $\chi^2$ | $(x-1)^2$ | $2\left(1 - \frac{\rho_\pi}{\rho_E}\right)$ |
| Total Variation | $\frac{1}{2}|x-1|$ | $\frac{1}{2} \cdot \text{sign}\left(1 - \frac{\rho_\pi}{\rho_E}\right)$ |

## A.4 OPTIMIZATION

In the adversarial imitation learning (AIL) framework, optimizing the $f$-divergence objective in Eq. 13 corresponds to the following iterative procedure that alternates between training a discriminator and updating the policy.

Assume a discriminator of the form $D(s,a) = \sigma(\ell(s,a))$, where $\ell(s,a)$ is a learned logit function and $\sigma(\cdot)$ denotes the sigmoid function. The goal of the discriminator is to distinguish between state-action pairs sampled from the expert occupancy measure $\rho_E$ and those induced by the current policy $\pi$, denoted $\rho$.

**Step 1: Discriminator Update.** The discriminator is trained by maximizing the binary cross-entropy objective:

$$D^*(s,a) = \arg\max_D \ \mathbb{E}_{(s,a)\sim\rho_E}\left[\log D(s,a)\right] + \mathbb{E}_{(s,a)\sim\rho}\left[\log(1 - D(s,a))\right]. \tag{36}$$

**Step 2: Policy Update.** The policy is then updated to maximize the expected return, where the reward is derived from the discriminator output via a function $r_{f/f\text{-var}}(s,a)$, which typically corresponds to a variational lower bound on the chosen $f$-divergence:

$$\pi^* = \arg\max_\pi \ \mathbb{E}_{(s,a)\sim\pi}\left[r_{f/f\text{-var}}(s,a)\right]. \tag{37}$$

where the choice of the reward assignment function $r_{f/f\text{-var}}$ depends on the $f$-divergence used.

Iterate between Steps 1 and 2 until convergence.

**Convergence.** As highlighted in (Ghasemipour et al., 2020), under the assumption that the discriminator is optimized to its optimum $D^*$ at each iteration, the overall procedure converges to a fixed point where the occupancy measure of the learned policy matches that of the expert, i.e., $\rho_\pi = \rho_E$.

## B PSEUDO-CODE

The complete pseudo-code for **f-AIL** is provided in Algorithm 1, while the LLM-guided evolutionary search is detailed in Algorithm 2.

---

**Algorithm 1 f-AIL**: Adversarial Imitation Learning via $f$-Divergence Minimization

---

**Require:** Expert trajectories $\tau_E \sim \pi_E$; initial policy, discriminator parameters $\theta_0$, $\phi_0$; $\lambda$ entropy coefficient, number of updates $T$

1: **for** iteration $i = 0, 1, \ldots, T$ **do**
2:     Sample trajectories $\tau_i \sim \pi_{\theta_i}$
3:     Update discriminator parameters $\phi_i \rightarrow \phi_{i+1}$ using the gradient:

$$\nabla_\phi \, \mathbb{E}_{(s,a)\sim\tau_E} \left[ \log D_\phi(s,a) \right] + \mathbb{E}_{(s,a)\sim\tau_i} \left[ \log(1 - D_\phi(s,a)) \right]$$

4:     Construct rewards using the discriminator logit: $r(s,a) \leftarrow r_f(\ell_{\phi_{i+1}}(s,a))$
5:     Compute advantage estimates $A(s,a)$ from trajectories $\tau_i$ using $r(s,a)$
6:     Update policy parameters $\theta_i \rightarrow \theta_{i+1}$ via PPO by optimizing

$$\nabla_\theta \mathbb{E}_{(s,a)\sim\tau_i} \left[ \min \left( r_t(\theta) A(s,a), \text{clip}(r_t(\theta), 1-\epsilon, 1+\epsilon) A(s,a) \right) \right] - \lambda \nabla_\theta \mathcal{H}(\pi_\theta)$$

      where $r_t(\theta) = \frac{\pi_\theta(a|s)}{\pi_{\theta_i}(a|s)}$ is the likelihood ratio, and $\mathcal{H}$ is the causal entropy.
7: **end for**
8: Sample trajectories $\tau \sim \pi_{\theta_T}$
9: Calculate Wasserstein distance between occupancy measures:

$$D_{\text{wasserstein}} = \mathcal{W}\big(\{(s,a)_{\pi_{\theta_T}}\}, \{(s,a)_{\pi_E}\}\big) \text{ where } (s,a)_{\pi_T} \sim \tau \text{ and } (s,a)_{\pi_E} \sim \tau_E$$

10: **return** $\pi_\theta, D_{\text{wasserstein}}$

---

**Algorithm 2** LLM-Guided Evolutionary Search

---

**Require:** Initial population of reward functions $\mathcal{P}_0 = \{r_f^{(j)}\}_{j=1}^P$, number of generations $G$, number of pairs $M$, candidates per pair $N$, selection size $K$

1: **for** generation $g = 1, \ldots, G$ **do**
2:     Randomly sample $M$ pairs $\{(r_f^{(p_1)}, r_f^{(p_2)})\}$ from current population $\mathcal{P}_{g-1}$
3:     *# Crossover Generation*
4:     **for** each pair $m = 1, \ldots, M$ **do**
5:         Use LLM to generate $N$ candidate reward functions $\{r_f^{(m,n)}\}_{n=1}^N$
6:     **end for**
7:     *# Fitness Evaluation*
8:     **for** each candidate $(m,n)$ **do**
9:         Run Algorithm 1 with reward assignment function $r_f^{(m,n)}$
10:         Obtain policy $\pi_{\theta_T}^{(m,n)}$ and Wasserstein distance $D_{\text{wasserstein}}^{(m,n)}$
11:     **end for**
12:     *# Selection*
13:     Select top $K$ candidates $\{r_f^*\}$ with least divergence from the union of the current population $\mathcal{P}_{g-1}$ and generated crossovers $\{r_f^{(m,n)}\}$:

$$\mathcal{P}_g \leftarrow \{r_f^*\}_{k=1}^K$$

14: **end for**
15: **return** Best reward function(s) $r_f^*$ and corresponding learned policies $\pi_{\theta_T}^*$

---

# C  PROMPT STRATEGY

## C.1  TEMPLATE

We use the following prompt to instruct the LLM to generate crossover RA functions:

---

**Prompt for Crossover Generation**

**Role: AI Research Assistant (Imitation Learning)**
**Overall Objective:** Collaborate to discover novel reward functions for Adversarial Imitation Learning (AIL) that improve **training stability** and **final policy performance**. Performance is measured by a performance score (higher is better).
**Background: Adversarial Imitation Learning Setting**
You have a policy $\pi$ and expert transitions $(s, a)$ stored in a dataset $D_E$. The typical learning loop involves:

1. Sampling transitions $(s, a)$ into a dataset $D_\pi$ using the current policy $\pi$.
2. Training a discriminator $D(s, a)$ to distinguish between expert transitions ($D_E$) and policy transitions ($D_\pi$) using a standard binary cross-entropy loss:

$$L = -\mathbb{E}_{(s,a) \sim D_E}[\log(D(s, a))] - \mathbb{E}_{(s,a) \sim D_\pi}[\log(1 - D(s, a))]$$

3. The discriminator's output logits, $l(s, a)$, approximate the log-density ratio:

$$l(s, a) \approx \log \frac{\rho^E(s, a)}{\rho^\pi(s, a)}.$$

4. Policy transitions $(s, a)$ in $D_\pi$ are assigned rewards based on these logits using a reward function $r(s, a) = f(l(s, a))$. Examples include:
   • **GAIL:** $r(s, a) = -\log(1 - D(s, a)) = \text{softplus}(l(s, a))$ (Smooth rectifier: near 0 for negative logits, linear for positive).
   • **AIRL:** $r(s, a) = \log D(s, a) - \log(1 - D(s, a)) = l(s, a)$ (Linear everywhere).
   • **FAIRL:** $r(s, a) = -l(s, a) \cdot \exp(l(s, a))$ (Rises from 0 to $1/e$ at $l = -1$, then drops sharply).
   • **LOGD:** $r(s, a) = \log D(s, a) = -\text{softplus}(-l(s, a))$ (Linear for negative logits, near 0 for positive).
5. The policy $\pi$ is updated using reinforcement learning (e.g., PPO, SAC) with these calculated rewards.
6. Steps 1–5 are repeated.

**Your Task in This Interaction:**
You will be presented with two reward functions, $f_1$ and $f_2$ (defined based on logits $l$), along with their observed performance. Your goal is to propose a *new* function (not the same as GAIL, AIRL, FAIRL, LOGD), $f_3$, that aims to perform better (higher score).
**Instructions:**

1. **Analyze $f_1$ and $f_2$:**
   • Consider their mathematical shapes and properties (e.g., monotonicity, bounds, smoothness).
   • Consider their behavior when the logits are near zero, positive, and negative. What signal do they provide?
   • Relate these properties to the provided performance data. Why might one function have performed better/worse?
2. **Design $f_3 = $ `reward_fn(logits)`:**
   • Based on your analysis, propose a *new* function $f_3$.
   • **Aim for diversity:** Propose a mix of novel functions and variations on the provided examples.
3. **Implementation Requirements:**
   • **Input:** `logits` (a JAX array).
   • **Output:** `reward` (a JAX array of the same shape).
   • **Language:** JAX.
   • **Function Name:** `reward_fn`.
   • **Clarity:** Ensure the code is clean, well-commented (if necessary), and easily extractable. Include necessary imports (`jax.numpy as jnp`, `jax.nn` etc.).
   • **Enclose in Code Block:** Use a code block with the language `python`.
   • **Jittable:** Ensure the function is jittable by JAX.

**Response Format:**

```python
import jax.numpy as jnp
# from jax import nn # Uncomment or add other imports if needed

def reward_fn(logits):
    """
    [Brief description of the function's logic/intent]
    """
    # [Your implementation here]
    reward = ...
    return reward
```

**Pair of Reward Functions:**
Function 1:

```
[Code]
```

Score: [score]
Function 2:

```
[Code]
```

Score: [score]

---

## C.2 GENERATED EXAMPLE:

Below is an example RA function generated by the LLM using the previously described prompt.

---

**Parent 1**

```
import jax
import jax.numpy as jnp
import jax.nn

def reward_fn(logits):
    """
    Softplus function, but clipped at a maximum value (e.g., 5.0).
    Prevents potentially destabilizing large positive rewards.
    r = min(softplus(logits), CLIP_VALUE)
    """
    CLIP_VALUE = 5.0
    return jnp.minimum(jax.nn.softplus(logits), CLIP_VALUE)
```

**Score: -10.0**

---

**Parent 2**

```
import jax.numpy as jnp

def reward_fn(logits):
    reward = -logits*jnp.exp(logits)
    return reward
```

**Score: -15.0**

---

**Generated Crossover**

```
import jax.numpy as jnp
import jax.nn

def reward_fn(logits):
    """
    Combines a smooth non-linear shaping with symmetric bounds:
    - Uses tanh scaled by softplus to keep rewards smooth and bounded.
    - For negative logits, rewards are near zero but smoothly increase toward small
    positive values, avoiding harsh penalties.
    - For positive logits, rewards grow but saturate due to tanh,
      mitigating exploding positive rewards that cause instability.
    Intuition:
    - This function gently encourages the policy towards regions where logits are positive,
      but heavily penalizing large values is avoided by the tanh saturation.
    - Provides smoother gradients near zero by multiplying softplus(logits) which behaves
    like ReLU.
    """
    # Softplus ensures smooth soft-rectifier for logits, avoiding zero-gradient dead zones.
    sp = jax.nn.softplus(logits)  # ~0 for large negative, linear for large positive

    # tanh bounds output between -1 and 1, adding mild negative rewards for
    # strongly negative logits
    # Scale tanh(logits) to control sharpness:
    scaled_tanh = jnp.tanh(logits / 2.0)

    # Combine signals multiplicatively:
    reward = sp * scaled_tanh

    return reward
```

---

## D HYPERPARAMETERS

The hyperparameters used across different components are summarized below. Table 4 contains the parameters for the LLM evolution process. The PPO hyperparameters used to train both the expert and the AIL policy are listed in Table 5, and the discriminator-specific settings are provided in Table 6. The hyperparameters for the A2C algorithm are reported in Table 7.

Table 4: LLM Evolution hyperparameters

| Parameter | Value |
|---|---|
| LLM Model | GPT-4.1-mini |
| Number of Generations | 10 |
| Number of Pairs | 20 |
| Candidates per Pair | 1 |
| Selection Size | 10 |
| Number of evaluation seeds | 16 |

Table 5: PPO hyperparameters for MinAtar and Brax

| Hyperparameter | MinAtar | Brax |
|---|---|---|
| Number of Environments | 64 | 2048 |
| Number of Env Steps | 128 | 10 |
| Total Timesteps | $1 \times 10^7$ | $5 \times 10^7$ |
| Number of Minibatches | 8 | 32 |
| Number of Epochs | 4 | 4 |
| Discount Factor | 0.99 | 0.99 |
| GAE $\lambda$ | 0.95 | 0.95 |
| PPO Clip | 0.2 | 0.2 |
| Value Function Coefficient | 0.5 | 0.5 |
| Entropy Coefficient | 0.01 | 0 |
| Max Gradient Norm | 0.5 | 0.5 |
| Layer Width | 64 | 256 |
| Number of Hidden Layers | 2 | 2 |
| Activation | relu | relu |
| LR | 0.005 | 0.0003 |
| Anneal LR | linear | none |
| Optimizer | adam | adam |

# E INDIVIDUAL TRAINING CURVES

Figure 9 presents the training curves for DAIL, and baseline methods. DAIL consistently outperforms the baselines across all evaluated environments, with the exception of AIRL outperforming DAIL on *Reacher*. Additionally, DAIL exhibits faster convergence even when final returns are comparable, underscoring the training stability introduced by its meta-learned reward assignment function.

# F DISCOVERED REWARD ASSIGNMENT FUNCTIONS

The top five reward assignment functions discovered are shown in Table 8. While the resulting functions are complex, they are partly composed of primitives found in the base population, such as $x$, $\log x$ and $e^x$ along with new ones such as $|x|$, $\min(x, y)$ and $\max(x, y)$.

# G RUNTIME AND COMPUTE USED

Our experiments were conducted using a mix of GPUs available on our compute cluster, including NVIDIA L40, A100, 3090, and H100 NVL. Each AIL evaluation involved training 16 agents in parallel, distributed equally across two GPUs. The wall-clock time for each evaluation is reported in Table 9.

# H COMPARISON WITH OPEN-ES

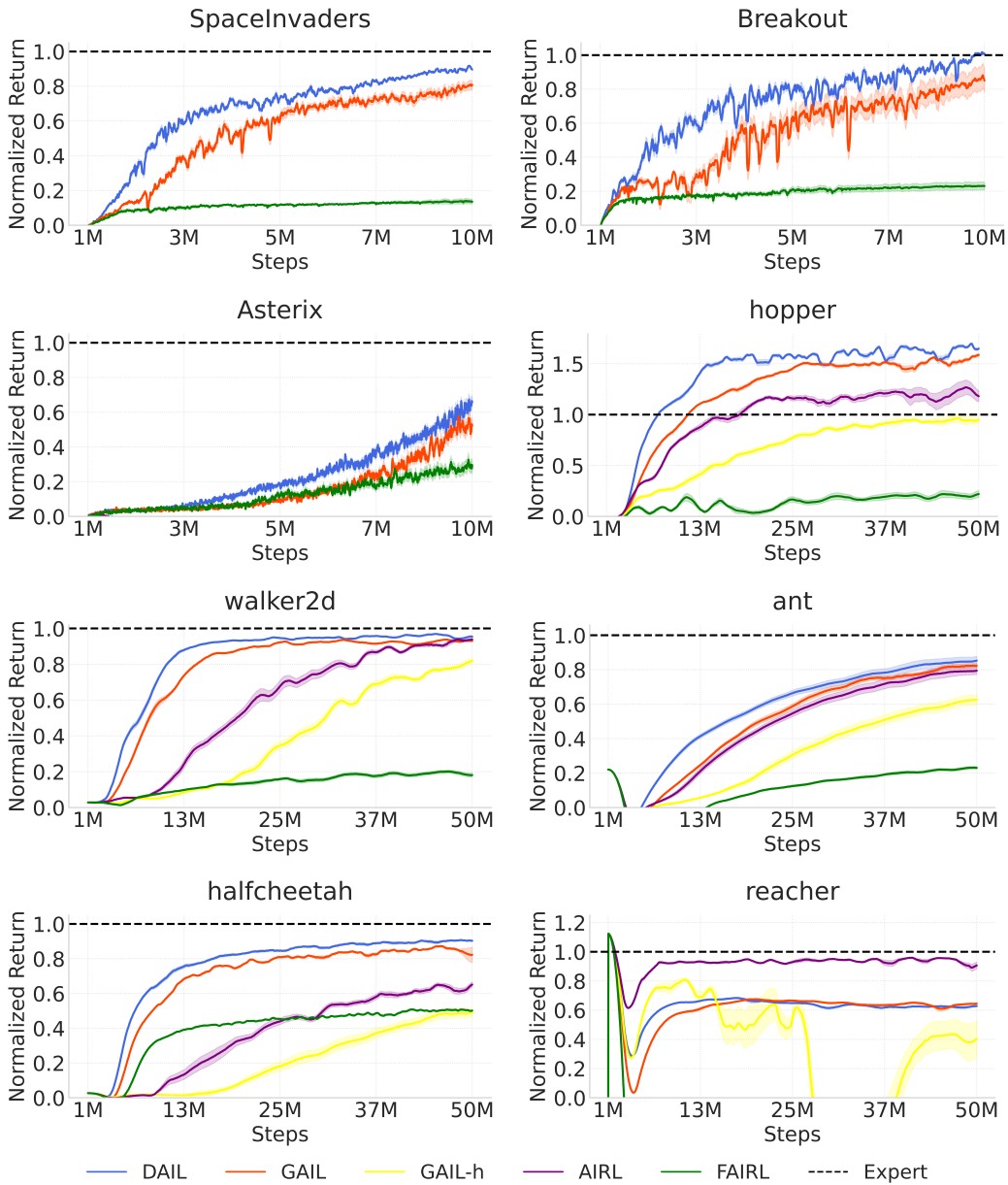

Figure 9: Mean normalized returns across all evaluated environments. DAIL consistently outperforms baseline methods, with the exception of AIRL on *Reacher*.

Table 6: Discriminator hyperparameters for Brax and Minatar.

| Hyperparameter | MinAtar | Brax |
|---|---|---|
| Layer Width | 64 | 128 |
| Number of Hidden Layers | 1 | 1 |
| Activation | relu | relu |
| Learning Rate (LR) | 0.0003 | 0.0003 |
| Gradient Penalty Weight | 0.1 | 1.0 |
| Number of Epochs | 1 | 1 |
| Number of Minibatches | 8 | 32 |
| Activation | relu | relu |
| Optimizer | adam | adam |

Table 7: A2C hyperparameters for MinAtar SpaceInvaders

| Hyperparameter | Value |
|---|---|
| Number of Environments | 64 |
| Number of Env Steps | 16 |
| Total Timesteps | $1 \times 10^7$ |
| Number of Minibatches | 8 |
| Discount Factor | 0.99 |
| GAE $\lambda$ | 0.95 |
| Value Function Coefficient | 5.0 |
| Entropy Coefficient | 0.01 |
| Max Gradient Norm | 10.0 |
| Layer Width | 64 |
| Number of Hidden Layers | 2 |
| Activation | relu |
| LR | 0.005 |
| Anneal LR | linear |
| Optimizer | adam |

We evaluate the contribution of the LLM in guiding the evolutionary search by comparing it to a widely used baseline for optimizing the outer-loop objective (Eq.3): OpenAI Evolution Strategies (ES) Salimans et al. (2017). OpenAI-ES is a black-box, gradient-free optimization algorithm known for its strong performance on similar classes of problems Goldie et al. (2024); Sapora et al. (2024). For a fair comparison, both methods are given the same computational budget (200 inner-loop evaluations). From Table 10, we observe that OPEN-ES underperforms DAIL both on the training environment (SpaceInvaders) and on the test environments overall, consistent with the findings reported by Goldie et al. (2025). Figure 10 visualizes the discovered RA function, which appears irregular and non-smooth, likely contributing to its inferior performance.

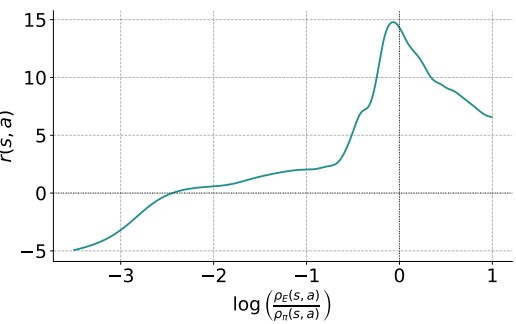

Figure 10: RA function generated by OpenES

# I  COMPARISON WITH ADDITIONAL DIVERGENCE BASED METHODS

We evaluate RA functions derived from alternative $f$-divergences, as summarized in Table 3, which has not been explored in prior work Ghasemipour et al. (2020); Orsini et al. (2021). We also compare

Table 8: Top 5 reward assignment functions discovered generated after evolution. Each function is expressed in terms of discriminator logits $l$.

| Environment | Reward Assignment Functions |
|---|---|
| SpaceInvaders-Run 1 | 1. $\sigma(l) \cdot 0.5 \cdot \big(\tanh(l) + 1\big)$ 

 2. $\min\left(1.5, \max\left(0, \begin{cases} 0.5 + 0.8\,l - \frac{\text{softplus}\big(1.5(-l-0.8)\big)}{1.5}, & l \leq -0.8 \\ 0.5 + 0.8\,l, & -0.8 < l < 0.8 \\ 0.5 + 0.8 \cdot 0.8 + \frac{\text{softplus}\big(1.5(l-0.8)\big)}{1.5}, & l \geq 0.8 \end{cases}\right)\right)$ 

 3. $\text{softplus}(l) \cdot \sigma(1.5l) + 0.5 \cdot \text{gelu}(l)$ 

 4. $\frac{l}{1+|l|} \cdot \sigma(3l) \cdot 0.5 \cdot \big(\tanh(l) + 1\big)$ 

 5. $0.5 \cdot \left(\frac{l}{1+|l|} + 1\right) \cdot \sigma(3l)$ |
| SpaceInvaders-Run 2 | 1. $\text{clip}\big(0.5 \cdot \big(\tanh(l) + 1\big) \cdot \text{softplus}(l) - 0.1\,\text{softplus}(-l),\, 0,\, \infty\big)$ 

 2. $\text{softplus}(l) \cdot \left(1 - \tanh^2\left(\frac{l-3}{1.5}\right)\right)$ 

 3. $\sigma(3l) \cdot \text{softplus}(l)$ 

 4. $\sigma\big(3(l-1)\big) \cdot \sigma\big(5(2.5 - |l|)\big)$ 

 5. $\text{clip}\left(\left(\left(\frac{l}{1+|l|} + 0.2\left(\frac{l}{1+|l|}\right)^3 + 0.3\right) \cdot \exp\big(-0.2\,\max(l,0)\big),\, 0,\, \infty\right)\right.$ |
| Breakout | 1. $0.5 \cdot \big(\tanh(1.5 \cdot l) + 1\big)$ 

 2. $\text{softplus}(l) \cdot \frac{l \cdot \sigma(l) + 1}{2}$ 

 3. $\big(\tanh(l) + 1\big) \cdot \sigma(l)$ 

 4. $\frac{\tanh(2.0 \cdot l)}{2.0} + 0.3 \cdot \text{softplus}(l) + 0.5$ 

 5. $(1 - \sigma(2.0 \cdot l)) \cdot 0.7 \cdot \text{softplus}(l) + \sigma(2.0 \cdot l) \cdot \big(\text{softplus}(l) + \text{clip}(0.3 \cdot l, -1, 2)\big)$ |

Table 9: All reported runtimes correspond to wall-clock time measured while training 16 agents in parallel on 2 H100 NVL GPUs.

| Environment | Wallclock time (s) |
|---|---|
| Ant | 346.53 |
| HalfCheetah | 1034.86 |
| Hopper | 585.97 |
| Walker2d | 701.22 |
| Reacher | 290.54 |
| Breakout | 55.07 |
| Asterix | 97.34 |
| SpaceInvaders | 52.57 |

DAIL to Wasserstein-GAIL (WGAIL) , which corresponds to replacing the discriminator objective

Table 10: Comparison between DAIL and OPEN-ES across MinAtar and Brax environments. Reported values denote mean $\pm$ standard error over evaluation runs. [†]Note that DAIL and OPEN-ES were meta-trained on MinAtar SpaceInvaders.

| Environment | DAIL | OPEN-ES |
|---|---|---|
| SpaceInvaders[†] | **0.90** $\pm$ **0.00** | 0.73 $\pm$ 0.05 |
| Asterix | 0.66 $\pm$ 0.03 | **1.27** $\pm$ **0.04** |
| Breakout | **1.01** $\pm$ **0.00** | 0.38 $\pm$ 0.10 |
| HalfCheetah | **0.90** $\pm$ **0.00** | 0.62 $\pm$ 0.02 |
| Walker2d | **0.95** $\pm$ **0.00** | 0.73 $\pm$ 0.02 |
| Hopper | **1.65** $\pm$ **0.01** | 1.09 $\pm$ 0.03 |
| Reacher | 0.63 $\pm$ 0.01 | **0.83** $\pm$ **0.01** |
| Ant | **0.85** $\pm$ **0.02** | 0.51 $\pm$ 0.03 |

Table 11: Comparison with additional divergence-based AIL methods across MinAtar and Brax environments. Reported values denote mean $\pm$ standard error.

| Environment | DAIL | Pearson | Sq-Hellinger | TV | WGAIL |
|---|---|---|---|---|---|
| Asterix | **0.66** $\pm$ **0.03** | -0.03 $\pm$ 0.00 | -0.02 $\pm$ 0.00 | -0.01 $\pm$ 0.00 | 0.52 $\pm$ 0.04 |
| Breakout | **1.01** $\pm$ **0.00** | -0.01 $\pm$ 0.00 | -0.01 $\pm$ 0.00 | -0.01 $\pm$ 0.00 | 0.91 $\pm$ 0.06 |
| Ant | **0.85** $\pm$ **0.02** | 0.57 $\pm$ 0.02 | 0.82 $\pm$ 0.03 | 0.19 $\pm$ 0.05 | 0.80 $\pm$ 0.02 |
| HalfCheetah | **0.90** $\pm$ **0.00** | 0.49 $\pm$ 0.04 | 0.86 $\pm$ 0.01 | -0.01 $\pm$ 0.01 | 0.87 $\pm$ 0.01 |
| Hopper | **1.65** $\pm$ **0.01** | 0.90 $\pm$ 0.11 | 1.49 $\pm$ 0.04 | 1.59 $\pm$ 0.04 | 1.50 $\pm$ 0.01 |
| Reacher | 0.63 $\pm$ 0.01 | 0.88 $\pm$ 0.05 | **0.94** $\pm$ **0.01** | 0.86 $\pm$ 0.07 | 0.42 $\pm$ 0.01 |
| Walker2d | 0.95 $\pm$ 0.00 | 0.54 $\pm$ 0.05 | **0.98** $\pm$ **0.00** | 0.91 $\pm$ 0.01 | 0.89 $\pm$ 0.01 |

(36) with:

$$D^*(s,a) = \arg \max_{D \in \{f : \|f\|_L \leq 1\}} \mathbb{E}_{(s,a)\sim\rho_E} [D(s,a)] - \mathbb{E}_{(s,a)\sim\rho} [D(s,a)] \qquad (38)$$

Note that we already employ gradient penalty as a regularizer, enforcing the discriminator to be 1-Lipschitz. After a hyperparameter sweep over the discriminator learning rate, we find that $3 \times 10^{-3}$ performs best for WGAIL. From Table 11, we observe that DAIL outperforms the divergence-based AIL baselines on 5 out of 7 test environments.

## J COMPARISON WITH NON-ADVERSARIAL METHODS

We additionally compare DAIL with non-adversarial methods, namely Behavior Cloning (BC; Pomerleau (1988)) and IQ-Learn (Garg et al., 2021), a state-of-the-art non-adversarial algorithm. For IQ-Learn, we use the official implementation provided by Garg et al. (2021). Results are reported in Table 12. DAIL consistently outperforms both BC and IQ-Learn. Interestingly, IQ-Learn fails to surpass BC in 3 of the 4 tested environments, a phenomenon also observed in prior work (Lai et al., 2024; Jain et al., 2024), highlighting stability challenges inherent in non-adversarial imitation learning methods as well.

Table 12: Comparison with **non-adversarial** imitation learning algorithms. Reported values are mean returns $\pm$ 95% confidence intervals across 4 random seeds for each environment.

| Method | Ant | HalfCheetah | Hopper | Walker2d |
|---|---|---|---|---|
| BC | 0.23 $\pm$ 0.04 | 0.09 $\pm$ 0.02 | 0.23 $\pm$ 0.09 | 0.04 $\pm$ 0.01 |
| IQ-learn | $-0.32$ $\pm$ 0.01 | $-0.02$ $\pm$ 0.00 | 0.02 $\pm$ 0.06 | 0.03 $\pm$ 0.01 |
| DAIL | 0.88 $\pm$ 0.06 | 0.90 $\pm$ 0.01 | 1.72 $\pm$ 0.04 | 0.97 $\pm$ 0.01 |

## K    LLM USAGE

LLMs have been used in the writing of this paper, primarily to refine the quality of the text through prompt-based polishing of author-written drafts.

