# OpenReview forum: "On Discovering Algorithms for Adversarial Imitation Learning"
_ICLR.cc/2026/Conference — ICLR 2026 Poster_

### Official Review · Reviewer_GF4H · 2025-10-17

**Soundness:** 2
**Presentation:** 3
**Contribution:** 2
**Rating:** 6
**Confidence:** 3

**Summary:**

This paper proposes Discovered Adversarial Imitation Learning (DAIL) that stabilizes the training process by using an LLM-guided framework to explore the Reward Assignment (RA) functions.

**Strengths:**

- This paper presents a search procedure that illustrate how the LLM-guided framework searches for the reward assignment functions
- Empirical experiments on MuJoCo and Minatar tasks have show the effectiveness of DAIL

**Weaknesses:**

- The paper primarily builds upon existing reward assignment strategies, and thus the level of novelty is limited.
- While experiments support the proposed idea, the paper lacks rigorous theoretical analysis to clarify to what extent DAIL improves stability or under what conditions the improvements are guaranteed.

**Questions:**

- Some of the text in Figure 2 is too small to read, which affects the clarity of the visualization. Improving figure readability would strengthen the presentation.
- There seems to be a contradiction regarding the reward assignment (RA) function: in Section 5.2, it appears to change dynamically, while the future work section mentions that the RA function in DAIL is static. Could the authors clarify whether the RA function is adaptive during training or fixed throughout?

---

> ### Author Response · Authors · 2025-11-22
>
> Thank you for taking the time to review our work and for your constructive feedback. We also note some misunderstandings, which we clarify below.
>
> > "The paper primarily builds upon existing reward assignment strategies, and thus the level of novelty is limited."
>
> We respectfully disagree that this choice limits the novelty of our approach. Initialization plays a crucial role in evolutionary methods and directly impacts sample efficiency~[5]; therefore, we ground the search with an initial set of well-established RA functions. We would like to clarify that **evolution is not restricted to the function family represented in the initial population**; these functions simply serve as starting candidates for the evolution.
>
> > While experiments support the proposed idea, the paper lacks rigorous theoretical analysis to clarify to what extent DAIL improves stability or under what conditions the improvements are guaranteed.
>
> We acknowledge that a deeper theoretical analysis of DAIL would further enrich our understanding. However, DAIL’s behavior is already structurally intuitive: with a fixed discriminator, it naturally drives policy improvement, as the reward increases monotonically with the log-ratio. It is also worth noting that many algorithms—for example, PPO—were adopted based on the strength of their empirical performance, with formal theoretical frameworks developed only later [7]. Moreover, meta-learning is fundamentally empirical in nature, as it is grounded in learning from data. Consequently, it is common in meta-learning for methods to demonstrate strong empirical results with limited theoretical guarantees  [8, 9, 10, 13]. In such settings, a mechanistic interpretation—such as the one provided in Section 6.3—offers meaningful and actionable insight. That said, we have begun preliminary theoretical analysis of function classes related to DAIL and are actively investigating their properties. However, we believe a thorough treatment of these results would extend beyond the empirical focus of the current work and would be better suited as a standalone follow-up paper.
>
> > "static v/s dynamic RA functions confusion"
>
> Note that after evolution, during the meta-testing stage, the RA function (DAIL) remains fixed throughout the adversarial imitation learning process (the inner loop). In the Future Work section, we discuss the possibility of adapting the RA function dynamically based on the training state of the imitation policy and discriminator.
>
> During meta-training (the outer loop), different RA functions are evolved—making this stage dynamic. However, each time an RA function is evaluated through the inner loop, it remains fixed for the duration of that evaluation. We hope this clarifies the distinction.
>
> ### References
> [1] Oh, J., Farquhar, G., Kemaev, I. et al. Discovering state-of-the-art reinforcement learning algorithms. Nature (2025).
> [2] Kuba, J. G., Schroeder de Witt, C., & Foerster, J. (2024). Mirror Learning: A Unifying Framework of Policy Optimisation. ICML 2022
> [3] Oh, J., Hessel, M., Czarnecki, W. M., Xu, Z., van Hasselt, H., Singh, S., & Silver, D. (2021). Discovering Reinforcement Learning Algorithms. NeurIPS 2020
> [4] Goldie, A. D., Lu, C., Jackson, M. T., Whiteson, S., & Foerster, J. N. (2025). Can Learned Optimization Make Reinforcement Learning Less Difficult? NeurIPS 2024
> [5] Lu C., Kuba J., Letcher A., Metz L., Schroeder de Witt C., and Foerster J. N.
> Discovered policy optimisation. NeurIPS 2022

---

### Official Review · Reviewer_JUMG · 2025-10-28

**Soundness:** 1
**Presentation:** 3
**Contribution:** 1
**Rating:** 2
**Confidence:** 5

**Summary:**

This paper proposes Discovered Adversarial Imitation Learning (DAIL), a meta-learning framework that aims to automatically discover reward assignment (RA) functions for adversarial imitation learning.
DAIL employs an LLM-guided evolutionary search to generate and evolve candidate RA functions expressed as executable code, optimizing them based on the Wasserstein distance between expert and policy occupancy measures.
Experiments on Brax and Minatar benchmarks show that DAIL achieves higher normalized returns and better alignment with expert behavior than existing AIL baselines.
The paper claims to offer the first meta-learned AIL algorithm and provides empirical analyses linking the discovered reward structure to more stable adversarial training dynamics.

**Strengths:**

1. *Novelty.* The paper presents a moderately original idea by framing RA discovery in AIL as a meta-learning problem and leveraging LLMs for evolutionary search over reward function space. Its originality stems primarily from the methodological integration of LLM-guided symbolic evolution into the imitation learning pipeline, rather than from a fundamentally new theoretical insight about AIL itself.

2. *Quality.* The paper is technically sound in its experimental setup and provides comprehensive empirical validation on both Brax and Minatar benchmarks, demonstrating consistent improvements in performance over standard baselines.

3. *Clarity.* The paper is clear and easy to follow.

4. *Significance.* The work contributes an interesting demonstration of how LLMs can aid in discovering algorithmic components in AIL. However, its broader impact on the IL/RL community is modest, as the main performance gains stem from empirical search rather than conceptual advances, and the underlying instability of AIL that is driven by the adversarial optimization process, remains unresolved.

**Weaknesses:**

The main weakness of this paper lies in its limited theoretical novelty and misattribution of the source of instability in AIL.

1. The idea of studying how RA functions affect learning dynamics is not new. Prior works [1,2,3] have already analyzed and benchmarked multiple RA designs under the f-divergence framework.

2. The proposed contribution of using an LLM-guided evolutionary search to discover new RA functions only offers a methodological novelty rather than a theoretical or algorithmic one. However, this approach does not address the primary cause of instability in AIL, which stems from the adversarial f-divergence min–max optimization dynamics rather than the form of the f-divergence [4]. As a result, the proposed method may yield empirically smoother rewards but cannot fundamentally stabilize training.

3. The discovered RA functions violate theoretical constraints that ensure valid divergence minimization, which undermines interpretability and convergence guarantees. The paper could be significantly strengthened by incorporating these mathematical constraints into the search process (e.g., by enforcing convexity) or by analyzing the discovered functions’ theoretical properties relative to valid divergences.

4. Empirically, while the experiments on Brax and Minatar demonstrate some improvements, the evaluation is limited to relatively simple continuous-control and grid-world tasks. The generality and robustness of DAIL would be more convincing if tested on high-dimensional or visual IL environments (e.g., Atari) and under different discriminator architectures or regularization strategies.

5. The paper attributes stability improvements solely to the discovered reward function, but it lacks ablation studies isolating the effect of LLM-guided search versus simpler baselines (e.g., random search, symbolic regression, or gradient-based meta-learning). Including such comparisons would clarify whether the LLM’s involvement meaningfully contributes beyond automated function optimization.

[1] Ke, Liyiming, et al. "Imitation learning as f-divergence minimization." International workshop on the algorithmic foundations of robotics. Cham: Springer International Publishing, 2020.
[2] Ghasemipour, Seyed Kamyar Seyed, Richard Zemel, and Shixiang Gu. "A divergence minimization perspective on imitation learning methods." Conference on robot learning. PMLR, 2020.
[3] Zhang, Xin, et al. "f-gail: Learning f-divergence for generative adversarial imitation learning." Advances in neural information processing systems 33 (2020): 12805-12815.
[4] Arjovsky, Martin, and Léon Bottou. "Towards principled methods for training generative adversarial networks." arXiv preprint arXiv:1701.04862 (2017).

**Questions:**

1. Why does the method use the Wasserstein distance as the evaluation objective while still training within the f-divergence–based AIL framework? If Wasserstein distance provides smoother and more stable gradients, why not directly optimize it instead of meta-learning new reward assignment functions?

2.How does the discovered reward function r theoretically relate to any valid divergence or distance measure?

3. Does the use of Wasserstein distance for evaluating fitness bias the search toward functions that implicitly approximate it?

4. How does DAIL compare in stability and performance to Wasserstein-based imitation methods [5, 6]?

5. Can the instability in AIL truly be mitigated by changing the RA, or does it fundamentally arise from the adversarial optimization dynamics?

6. Would replacing the adversarial discriminator with a 1-Lipschitz critic (as in WGANs) render the entire reward-assignment discovery unnecessary?

[5] Zhang, Ming, et al. "Wasserstein distance guided adversarial imitation learning with reward shape exploration." 2020 IEEE 9th Data Driven Control and Learning Systems Conference (DDCLS). IEEE, 2020.
[6] Dadashi, Robert, et al. "Primal wasserstein imitation learning." arXiv preprint arXiv:2006.04678 (2020).

---

> ### Author Response · Authors · 2025-11-22
>
> We sincerely appreciate your constructive and detailed feedback. Your suggestions—particularly the proposed additional experiments—proved invaluable and helped highlight the strengths of our method. We also realize that some aspects of the paper may have been misunderstood, and we address these clarifications below along with direct responses to your comments.
>
> > "The idea of studying how RA functions affect learning dynamics is not new."
>
> We have acknowledged in the Related Work section (Line 103) that we are not the first to benchmark different f-divergence–based RA functions, although we did miss the citations [1,3]—thank you for pointing this out; we have now added it. We also revised Section 4, where we discuss the impact of RA functions, to explicitly note that this was first introduced in [2].
>
> Importantly, the main contribution of our work lies in meta-learning data-driven RA functions, rather than relying on human-designed ones, which leads to the **first meta-learned AIL algorithm**.
>
> > "adversarial IL is fundamentally instable"
>
> We respectfully disagree with the assertions that (a) adversarial training cannot fundamentally be stabilized and (b) the choice of f-divergence cannot contribute significantly to such stabilization. As discussed in [4], the main sources of instability in GANs are (a) discriminator overfitting (Section 2.1) and (b) the choice of cost function (Section 2.2), which is analogous to the RA function in our setting. As noted in Section 1 (Line 75), addressing these two factors leads to more stable training—indeed, [14, 15] demonstrate that improved discriminator loss functions enhance stability—and our work primarily focuses on the latter. The effect of RA functions on training stability has been emphasised in prior work [1, 2, 3] as well. Furthermore, **DAIL achieved a mean normalized reward of 95% across the tested environments, providing empirical evidence of its stable training**. This is further supported by the analysis in Figure 6 (left), which shows that DAIL exhibits training dynamics similar to those observed when using the ground-truth reward function.
>
> > "limited theoretical analyis"
>
> We acknowledge that a deeper theoretical analysis of DAIL would further enrich our understanding. However, DAIL’s behavior is already structurally intuitive: with a fixed discriminator, it naturally drives policy improvement, as the reward increases monotonically with the log-ratio. It is also worth noting that many algorithms—for example, PPO—were adopted based on the strength of their empirical performance, with formal theoretical frameworks developed only later [7]. Moreover, meta-learning is fundamentally empirical in nature, as it is grounded in learning from data. Consequently, it is common in meta-learning for methods to demonstrate strong empirical results with limited theoretical guarantees  [8, 9, 10, 13]. In such settings, a mechanistic interpretation—such as the one provided in Section 6.3—offers meaningful and actionable insight. That said, we have begun preliminary theoretical analysis of function classes related to DAIL and are actively investigating their properties. However, we believe a thorough treatment of these results would extend beyond the empirical focus of the current work and would be better suited as a standalone follow-up paper.
>
> > "incorporating mathematical constraints into the search"
>
> We place no convexity constraints on the RA function to allow the evolution process to discover solutions beyond the f-divergence family. If the optimal solution does correspond to an f-divergence, we expect the evolution to naturally produce convex functions.
>
> > "experiments on grid world tasks"
>
> We clarify that we **do not use gridworld environments** in any of our experiments. Meta-training on full Atari is not feasible for us because the environments are not JAX-based and therefore cannot be GPU-accelerated, placing them outside our computational budget. However, MinAtar provides a visually based, miniaturized version of the Atari suite, so our experiments do include visual environments.
>
> > "testing across different discriminator regularizers"
>
> We assess DAIL’s generalization performance under various discriminator regularization strategies proposed by Orsinin et al (2021), including weight decay, an additional entropy bonus, and spectral normalization of the discriminator weights. We also consider the case without any regularization. The results, summarized in Table 2 (main paper), show that DAIL outperforms GAIL in 3 out of 5 regularization regimes.

---

> ### Author Response · Authors · 2025-11-22
>
> > "effect of LLM-guided search vs simpler baselines"
>
> We employ LLMs for optimization since they are sample-efficient, interpretable, and capable of discovering generalizable solutions in similar settings (Line 260). Nevertheless, we evaluate the contribution of the LLM in this setting by comparing it to a widely used baseline: OpenAI Evolution Strategies (ES) [11]. For a fair comparison, both methods are given the same computational budget (200 inner-loop evaluations). From Table 10, we observe that OPEN-ES underperforms DAIL both in the training environment (SpaceInvaders) and in the test environments overall, consistent with the findings reported by [12]. Figure 10 visualises the discovered RA function, which appears irregular and non-smooth—likely contributing to its inferior performance. Please refer to Appendix H for more information on this experiment.
>
> > "why not directly optimize it instead of meta-learning new reward assignment functions?"
>
> The primary objective of divergence-minimization AIL is to match the expert’s occupancy measure, i.e.,\rho_{\pi} = \rho_{E} \; \forall (s,a), which is the unique minimizer for most divergences (e.g., KL, JS, Wasserstein). We adopt the Wasserstein distance due to its favorable behavior when \rho_{\pi} \neq \rho_{E} it is symmetric and does not require overlapping support over all (s,a), unlike KL or JS divergences. Consequently, it provides a more informative and stable evaluation signal.
>
> Directly optimizing the Wasserstein distance is **challenging and computationally expensive**. For instance, the approach proposed in [5] is limited to Euclidean cost matrices, while [6] relies on a greedy approximation (to reduce computational cost). Moreover, it is not straightforward to meta-learn an RA function that can be applied across environments with different state–action spaces. Hence, we use an f-divergence–inspired framework for RA function discovery, leveraging its density-ratio formulation to ensure applicability across diverse environments.
>
>
> > "Does the use of Wasserstein distance for evaluating fitness bias the search toward functions that implicitly approximate it?"
>
> We believe the Wasserstein distance provides a strong inductive bias, due to its robust estimation properties (Line~213). It also offers several favorable characteristics: (a) it defines a true metric, (b) it is symmetric, and (c) it reflects meaningful changes in the underlying metric space. In contrast, asymmetric divergences such as KL can lead to undesirable behaviors like mode\text{-}seeking and often require additional assumptions, including overlapping support. While alternative fitness measures may exist, exploring them falls outside the scope of this work and is left for future investigation.
>
> > "How does DAIL compare in stability and performance to Wasserstein-based imitation methods [5, 6]?"
>
> We already employ a 1-Lipschitz discriminator in our main experiments through gradient penalty regularization. WGAIL [5] modifies the discriminator objective by replacing the binary cross-entropy loss with a mean-difference loss, as outlined in Appendix I. In our experiments, WGAIL consistently lags behind DAIL in all environments (Table 11); additional details are provided in Appendix I.
>
> ### References:
> [7] Kuba, J. G., Schroeder de Witt, C., & Foerster, J. (2024). Mirror Learning: A Unifying Framework of Policy Optimisation. ICML 2022
> [8] Oh, J., Hessel, M., Czarnecki, W. M., Xu, Z., van Hasselt, H., Singh, S., & Silver, D. (2021). Discovering Reinforcement Learning Algorithms. NeurIPS 2020
> [9] Oh, J., Farquhar, G., Kemaev, I. et al. Discovering state-of-the-art reinforcement learning algorithms. Nature (2025).
> [10] Goldie, A. D., Lu, C., Jackson, M. T., Whiteson, S., & Foerster, J. N. (2025). Can Learned Optimization Make Reinforcement Learning Less Difficult? NeurIPS 2024
> [11] Salimans, T., Ho, J., Chen, X., Sidor, S., & Sutskever, I. (2017). Evolution Strategies as a Scalable Alternative to Reinforcement Learning. arXiv preprint arXiv:1703.03864.
> [12] Goldie, A. D., Wang, Z., Cohen, J., Foerster, J. N., & Whiteson, S. (2025). How Should We Meta-Learn Reinforcement Learning Algorithms? RLC 2025
> [13] Ramachandran, P., Zoph, B., & Le, Q. V. (2017). Searching for activation functions. arXiv preprint arXiv:1710.05941.
> [14] Luo, T., Pearce, T., Chen, H., Chen, J., & Zhu, J. (2024). C-GAIL: Stabilizing generative adversarial imitation learning with control theory. Advances in Neural Information Processing Systems, 37, 29464-29488.
> [15] Lai, C. M., Wang, H. C., Hsieh, P. C., Wang, F., Chen, M. H., & Sun, S. H. (2024). Diffusion-reward adversarial imitation learning. Advances in Neural Information Processing Systems, 37, 95456-95487.

---

### Official Review · Reviewer_6x5G · 2025-10-31

**Soundness:** 4
**Presentation:** 3
**Contribution:** 4
**Rating:** 6
**Confidence:** 4

**Summary:**

This paper addresses the long-overlooked reward assignment (RA) function design in Adversarial Imitation Learning (AIL) by proposing an LLM-guided evolutionary search framework. The authors formalize RA function discovery as a meta-learning problem optimizing Wasserstein distance, conducting the search on MinAtar SpaceInvaders to discover DAIL's RA function: r(x)=0.5·sigmoid(x)·[tanh(x)+1]. Experiments demonstrate that DAIL outperforms GAIL, AIRL, FAIRL, and other baselines across multiple unseen environments in both Brax and MinAtar suites. The authors further analyze policy entropy and log-ratio distributions to elucidate the mechanisms behind DAIL's enhanced training stability.

**Strengths:**

1. The AIL literature has predominantly focused on discriminator stabilization (Wang et al. 2024; Luo et al. 2024), while RA function design has been consistently overlooked. This paper is the first to systematically treat RA functions as an independent research subject, filling a significant void. More importantly, the authors shift the RA design paradigm from theory-driven (deriving from f-divergences) to data-driven (directly optimizing performance metrics), offering a fresh methodological perspective for AIL algorithm design. The discovered DAIL function is remarkably concise (one line of code) and easily integrable into existing AIL frameworks, demonstrating high practical value.

2. The choice of Wasserstein distance as the fitness function is natural, given its theoretical robustness in measuring distributional discrepancies. The LLM-guided evolutionary search organically combines the interpretability of code representation with search efficiency, providing more targeted exploration compared to pure random search. The initial population includes classical methods (GAIL, AIRL, FAIRL, GAIL-heuristic), ensuring high-quality starting points—a design choice that reflects the authors' deep domain understanding. Overall, from problem formalization to search strategy selection, the design demonstrates sound intuition.

3.  The training dynamics analysis in Figure 6, examining policy entropy and log-ratio distributions, clearly reveals why DAIL outperforms baselines: DAIL saturates to near-zero in the log-ratio < -1.8 region, effectively filtering noisy signals corresponding to random or low-quality behaviors, thereby achieving more stable training. This mechanistic explanation is not only data-supported (policy entropy converges faster to levels approaching PPO with true rewards) but also holds consistently across multiple environments (though space constraints limit presentation to selected examples), strengthening the credibility of the conclusions. The ablation study in Figure 7 further confirms the necessity of combining sigmoid and tanh, showing that using either function alone cannot match the combined effect.

**Weaknesses:**

1. The lack of theoretical guarantees is the most serious issue in this work. DAIL's RA function does not correspond to any known f-divergence, and thus lacks convergence guarantees. While the paper acknowledges this limitation (Lines 247-251), it merely justifies the approach with "empirical feedback" and "robust generalization" without providing any alternative theoretical analysis. In contrast, GAIL, AIRL, and FAIRL all have formal proofs of convergence to ρ_π = ρ_E (Ghasemipour et al. 2020). This deficiency means we cannot predict under what conditions DAIL will be effective or when it might fail, nor can it guide future RA function design. For an algorithmic paper, relying entirely on empirical results without theoretical foundation is a critical weakness. Even if rigorous convergence proofs are intractable, the authors should at least provide intuitive analysis under relaxed assumptions (e.g., bounded MDPs, Lipschitz continuous policy classes, approximately optimal discriminators) or explicitly discuss potential failure modes and applicability boundaries.

2. The evidence for generalization is fundamentally insufficient. The paper's core contribution claims to have "discovered a generalizable RA function," but the evidence supporting this claim has serious problems. According to Section 6.1 (Lines 347-353), the authors conduct the search on only one environment—MinAtar SpaceInvaders—and then test on the other 7 environments (5 Brax + 2 other MinAtar). This raises two critical concerns: First, SpaceInvaders is an Atari game with discrete action space, while Brax environments involve continuous control tasks with entirely different dynamics. That an RA function discovered on SpaceInvaders performs well on Brax could be a coincidence (SpaceInvaders happens to be a "good" meta-training environment) rather than evidence of DAIL's true cross-domain generalization capability. Second, different task types (e.g., sparse vs. dense reward environments, or varying state space dimensions) may require RA functions with different characteristics, yet the paper does not explore this at all.

3. The missing critical experiments make the generalization claims unconvincing: We would expect to see searches conducted independently on at least 2-3 different environments (e.g., search on Ant then test on Walker/HalfCheetah; search on Breakout then test on Asterix/SpaceInvaders), followed by comparison of whether the discovered RA functions are consistent. Only if multiple independent searches converge to similar functional forms would that constitute genuine evidence of generalization. The current experimental design at best demonstrates that "SpaceInvaders is a good meta-training task" but cannot establish "DAIL's generalization capability." The performance improvements shown in Figure 4 may merely reflect the peculiarities of SpaceInvaders rather than the method's universality.

4. Several writing and presentation details could be improved. Upon verification, the paper is largely consistent in notation usage (primarily using rfr_f
rf​ and ℓ\ell
ℓ), but the statement "offers less flexibility" on Line 85 is overly vague and requires more specific explanation of which aspects of non-adversarial methods lack flexibility. The title of Appendix A.2 presents potential confusion: since the paper uniformly uses the form Df(ρE∣∣ρπ)D_f(\rho_E || \rho_\pi)
Df​(ρE​∣∣ρπ​), whether FAIRL should be labeled as "Forward KL viewed as reverse" or simply "Forward KL" depends on the reader's interpretive perspective. The authors should clarify this explicitly in the title or main text to avoid misunderstanding.

**Questions:**

1. If the search were conducted independently on Brax environments such as Ant or Walker, would the resulting RA functions be similar in form to the DAIL function discovered on SpaceInvaders? How large would the performance differences be? This experiment is crucial for validating the method's true generalization capability. Relatedly, are the performance differences among the top-5 functions shown in Table 6 statistically significant?

2. Although DAIL does not strictly correspond to any f-divergence, could the authors provide intuitive analysis or informal reasoning about convergence under relaxed assumptions (e.g., small learning rates, approximately optimal discriminators, bounded state spaces)? Even without proving convergence, could the authors explicitly discuss the types of environments or conditions where DAIL might fail (e.g., highly non-stationary environments, tasks with extremely sparse rewards)?

3. If multiple independent searches were conducted (using different random seeds to initialize the LLM or population), how high would the overlap be among the top-K functions? This would reflect the stability of the search process. Additionally, could the authors report the complete computational cost (total GPU hours, rather than the per-evaluation time shown in Table 7)? Given 200 candidate functions, 10 generations of evolution, and each evaluation requiring training 16 agents to convergence, the total cost could be substantial. Clarifying this is important for assessing the method's practicality.

4. Why does IQ-Learn's performance in Table 8 significantly underperform its original paper's reports (even failing to beat simple BC in multiple environments)? Has the implementation correctness been verified? How were the hyperparameters tuned? This anomalous result weakens the credibility of comparisons with non-adversarial methods. Additionally, since the paper uses Wasserstein distance as both the fitness function and evaluation metric, could the authors specify what cost matrix is used (Euclidean distance? or other?)

---

> ### Author Response · Authors · 2025-11-22
>
> Thank you for the constructive feedback and the insightful questions regarding the stability of the evolution process. Your comments prompted us to clarify key points and conduct additional experiments to further evaluate and confirm the stability of the evoltuion, ultimately strengthening our paper. We also address several of your other questions below and appreciate the opportunity to clarify these points.
>
> > "generalization results are insufficient, could be a coincidence"
>
> We respectfully disagree that the evidence for generalization is insufficient. As you correctly stated, a single RA function (DAIL) demonstrates strong performance across 7 unseen environments that differ substantially from the meta-training environment, which by itself indicates generalization. The fact that DAIL can be applied across these diverse environments—each with distinct state–action spaces—is due to the RA function operating on the log-density ratio, which is environment-agnostic. This consistent performance highlights the robustness of the discovered RA function to variations in environment dynamics.
>
> Second, we deliberately selected Space Invaders for meta-training (as noted in Line 319), as it has been shown in prior work to facilitate the discovery of generalizable algorithms. In meta-learning, as in any machine learning setting, the quality and diversity of the training data are critical in determining the learned solution~[1] (Line 58); therefore, selecting an appropriate training environment is essential. Nevertheless, we also performed the evolutionary search on the MinAtar Breakout environment and found that DAIL emerged among the top five RA functions after evolution, speaking to the generality of the method.
>
> > "different environments have different optimal RA functions"
>
> Specializing the RA function to individual tasks is an interesting direction for future work. However, this study focuses on optimizing a general RA function that performs well across multiple tasks collectively. That said, our method can be readily adapted for task-specific specialization by treating the meta-test environment as the meta-train environment. The success of meta-training on MinAtar SpaceInvaders provides empirical evidence that this approach is feasible.
>
> > "several writing details, forward-reverse KL confusion"
>
> For the first part of your question, can we confirm if you’re referring to Line 113 - “they provide limited flexibility in accommodating scenarios such as state-only demonstrations (Torabi et al., 2018; Jain et al., 2024) and reward shaping (Sapora et al.,2024)”.
>
> Thank you for pointing this out. We have removed the note (Line 900) to avoid potential confusion and to maintain consistency throughout the paper.
>
> > "are the performance differences among the top-5 functions shown in Table 6 statistically significant?"
>
> We find that the differences among the top-k candidates in the final generation are not statistically significant, likely because they share very similar structural forms, as shown in Figure~8.
>
> > "could the authors provide intuitive analysis or informal reasoning about convergence under relaxed assumptions (e.g., small learning rates, approximately optimal discriminators, bounded state spaces)?"
>
> We acknowledge that a deeper theoretical analysis of DAIL would further enrich our understanding. However, DAIL’s behavior is already structurally intuitive: with a fixed discriminator, it naturally drives policy improvement, as the reward increases monotonically with the log-ratio. It is also worth noting that many algorithms—for example, PPO—were adopted based on the strength of their empirical performance, with formal theoretical frameworks developed only later [7]. Moreover, meta-learning is fundamentally empirical in nature, as it is grounded in learning from data. Consequently, it is common in meta-learning for methods to demonstrate strong empirical results with limited theoretical guarantees  [8, 9, 10, 13]. In such settings, a mechanistic interpretation—such as the one provided in Section 6.3—offers meaningful and actionable insight. That said, we have begun preliminary theoretical analysis of function classes related to DAIL and are actively investigating their properties. However, we believe a thorough treatment of these results would extend beyond the empirical focus of the current work and would be better suited as a standalone follow-up paper.

---

> > ### Author Response · Authors · 2025-11-22
> >
> > > "if multilple independant searches were conducted, and on a different environment"
> >
> > To assess the stability of the evolutionary process, we conduct an additional independent evolutionary run on the \textit{Minatar SpaceInvaders} environment. The top-5 RA functions discovered in both runs are presented in Table 8 and the top-3 are plotted in Figure 8. We see that the $6$ evolved RA functions exhibit highly similar structures. Further, they maintain informative gradients within the range $[-1, 0]$ and saturate rapidly thereafter, consistent with our analysis in Section 6.3. Furthermore, we conduct an evolutionary run in the \textit{MinAtar Breakout} environment and find that DAIL emerges among the top-5 RA functions in the final generation, demonstrating both the stability of the evolutionary process and the effectiveness of DAIL. Please refer to Section 6.4 and Appendix F for more information.
> >
> > > "GPU hours consumed"
> >
> > We note that the training times reported in Table 7 include the parallel training of 16 agents, enabled by JAX-based GPU-accelerated training pipeline. To clarify, we use 20 samples per generation over 10 generations, resulting in a total of 200 generated samples. As stated in Line 352, the meta-training process on MinAtar SpaceInvaders takes approximately 3 hours (wall-clock time), and 6 GPU hours (we use 2 GPUs).
> >
> > > "IQ-Learn results"
> >
> > We use the official implementation of IQ-Learn and adopt the default hyperparameters provided by the authors in our experiments, as Brax is built on the same MuJoCo physics engine. Because this implementation is not optimized for GPU-accelerated hardware, extensive hyperparameter tuning was computationally expensive. We also acknowledge that similar results have been reported in other studies (see Appendix J).
> >
> > > "Cost matrix used"
> >
> > We use the euclidean cost matrix to calculate the Wasserstein distance.
> >
> > ### References
> > [1] Oh, J., Farquhar, G., Kemaev, I. et al. Discovering state-of-the-art reinforcement learning algorithms. Nature (2025).
> > [2] Kuba, J. G., Schroeder de Witt, C., & Foerster, J. (2024). Mirror Learning: A Unifying Framework of Policy Optimisation. ICML 2022
> > [3] Oh, J., Hessel, M., Czarnecki, W. M., Xu, Z., van Hasselt, H., Singh, S., & Silver, D. (2021). Discovering Reinforcement Learning Algorithms. NeurIPS 2020
> > [4] Goldie, A. D., Lu, C., Jackson, M. T., Whiteson, S., & Foerster, J. N. (2025). Can Learned Optimization Make Reinforcement Learning Less Difficult? NeurIPS 2024
> > [5] Finn, C., Abbeel, P., & Levine, S. (2017, July). Model-agnostic meta-learning for fast adaptation of deep networks. In International conference on machine learning (pp. 1126-1135). PMLR.

---

### Official Review · Reviewer_5SFa · 2025-11-01

**Soundness:** 4
**Presentation:** 3
**Contribution:** 3
**Rating:** 8
**Confidence:** 3

**Summary:**

This paper focuses on discovering useful reward functions for adversarial imitation learning, with the goal of improving downstream performance of agents beyond that of hand-designed IL baselines that rely on f-divergence minimization. The paper proposes DAIL, the first meta-learned AIL algorithm, which outperforms standard AIL algorithms on a variety of tasks in continuous control (Brax) and in Atari (MinAtar).

DAIL formulates the meta-learning problem as minimizing the Wasserstein distance between the expert distribution induced by the dataset and the optimal policy with respect to the meta-learned reward function. The meta-learning approach is evolutionary search, which is simulated by an LLM that takes in two "parent" reward functions and aims to produce a "child" reward function that combines desired properties of both parents, making it more suitable for downstream IRL. Once a said reward function is given, standard AIL proceeds, and the Wasserstein distance between the final learned policy and the expert distribution is estimated, which is signal for the meta-learner.

**Strengths:**

The paper is very cleanly written, and even as a person who doesn't have much of a meta-learning or metagradient RL background, I was easily able to understand the problem and semi-verify that the approach was reasonable. The approach is also fairly novel -- as the paper claims, and as I have verified, there has been no prior work that has attempted to meta-learn a reward function via standard meta-gradient or evolutionary search approaches in AIL.

The results in the paper are also quite strong, improving upon standard baselines such as GAIL and AIRL somewhat considerably based on the probability of improvement plot in Figure 5, meaning that the meta-learning approach is yielding strong reward functions in practice.

**Weaknesses:**

For the scope of this paper, I don't really know if this should count as a weakness, but I would have liked to see some additional baseline comparison to other AIL algorithms beyond AIRL and GAIL. In particular, there are many possible f-divergences or IPMs that could be minimized to obtain a strong adversarial IL algorithm in practice, including the Wasserstein distance (e.g. see Wasserstein GAN [1]). This is in particular a useful baseline because the meta-learning objective is aiming to minimize the Wasserstein distance between the expert policy and the learned policy, which a Wasserstein AIL objective building on top of the Wasserstein GAN would in theory do. This has also been done in other work, such as the hybrid IRL paper [2].

In addition, it may have been a good idea in my opinion to see how the meta-learned reward function translated to much more complicated tasks than those of Brax and MinAtar. This was done in the Atari suite by Oh et. al. [3] when they discovered RL update rules, so maybe from an experimental standpoint it would be a good idea here. However, this is not a huge weakness in my eyes as Brax and other MinAtar environments are already much different than MinAtar SpaceInvaders, which was used for meta-training. Additionally, maybe some work could've been done focusing on if meta-learning the reward function across more environments would've led to a more robust RL algorithm.

[1]: Ishaan Gulrajani, Faruk Ahmed, Martin Arjovsky, Vincent Dumoulin, and Aaron Courville: Improved Training of Wasserstein GANs (NIPS 2017)
[2] Juntao Ren, Gokul Swamy, Zhiwei Steven Wu, J. Andrew Bagnell, and Sanjiban Choudhury: Hybrid Inverse Reinforcement Learning (ICML 2024)
[3] Junhyuk Oh, Matteo Hessel, Wojciech M. Czarnecki, Zhongwen Xu, Hado van Hasselt, Satinder Singh, and David Silver: Discovering Reinforcement Learning Algorithms (NeurIPS 2020).

**Questions:**

I don't have any major questions right now -- I think the paper is pretty clear and I think I understood the main points reasonably clearly.

---

> ### Author Response · Authors · 2025-11-22
>
> Thank you for your positive comments on both the method and the paper’s writing. We also appreciate your suggestion to compare against additional non-standard f-divergences and Wasserstein AIL. We have conducted these comparisons (see below).
>
> Other reviewers have also requested additional experiments, and to which we have found additional empirical evidence that further highlight the strength of DAIL. We hope these results would reinforce your confidence in our work. Thanks again for your help and feedback!
>
> ### **Addressing comments**
>
> > "additional f-divergences and Wasserstein AIL baselines"
>
> Thank you for your suggestion regarding additional baselines. In response, we have expanded our experimental evaluation to include three new $f$-divergence--based RA functions as well as WGAIL[4], which minimizes the Wasserstein-1 objective. As shown in the updated Table 3, detailed further in Appendix~I and Table 11, DAIL outperforms all of these additional baselines across the evaluated environments. This experiment suggestion is crucial for strengthening the effectiveness of our method, and we thank the reviewer for it.
>
> > "experiments on Atari suite + meta-training on multiple envs"
>
> We agree that evaluating DAIL’s generalization to more complex environments, such as the Atari and Procgen suites, is a natural next step. Additionally, prior work in meta-learning [3, 5] has shown that training across a diverse set of environments leads to more robust algorithms—for instance, [5] introduced the Disco57 and Disco103 suites comprising 57 and 103 environments, respectively. Exploring the discovery of AIL algorithms meta-trained on such large-scale environment sets is indeed compelling, but currently beyond our computational budget. We have included both the above points in the Future Work section.
>
> ### Citations
>
> [4] Zhang, Ming, et al. "Wasserstein distance guided adversarial imitation learning with reward shape exploration."
> [5] Oh, J., Farquhar, G., Kemaev, I. et al. Discovering state-of-the-art reinforcement learning algorithms. Nature (2025).

---

### Author Response · Authors · 2025-11-22

We thank all reviewers for their time, careful reading, and detailed feedback. The insightful questions and requested experiments have helped further strengthen this work, and we sincerely appreciate their contributions. During the discussion period, we aim to address each comment thoroughly and clarify any remaining concerns.

We begin by highlighting the main strengths identified by the reviewers:

### Key Strengths Highlighted by the Reviewers

- Reviewers 5SFa and JUMG praised the clarity and quality of the writing, and all reviewers agreed that the method is technically sound.
- All reviewers remarked on the strength of our empirical results, noting that DAIL consistently outperforms existing baselines across seven unseen environments, a promising outcome for adversarial imitation learning.
- Reviewer 6x5G appreciated our analysis of DAIL’s interpretability and the underlying factors contributing to its superior performance.

Next, we summarize the primary contributions of our work:

### Primary Contributions (including additional results added in response to reviewer feedback)
- We introduce the problem of meta-learning reward-assignment (RA) functions for more stable AIL training. Optimizing this leads to DAIL, the **first meta-learned AIL algorithm**.
- We present extensive experiments demonstrating DAIL’s generalization across unseen environments, policy optimization algorithms, and discriminator-regularization strategies.
- We provide a detailed mechanistic interpretation that explains why DAIL outperforms prior methods.
- We analyze the stability of the evolutionary process and provide evidence for the reproducibility of DAIL.

### Questions
We provide detailed, reviewer-specific responses to all questions / concerned raised. If further clarification is needed, we are more than happy to discuss.

---

> ### Author Response · Authors · 2025-11-23
>
> ### Results Summary
> We try to summarize the new experimental results below for convenience. For other non-tabular (i.e., figures) results and detailed written analysis, we provide brief comments and we strongly recommend the reviewers to check out the updated manuscript instead for a clearer view. The newly added material is colored $\color{teal}{\text{teal}}$ in the paper.
>
> ### **Table 8 and Figure 8: Our Evolutionary Framework is Stable!**
> **Summary**: We tested our evolutionary framework for stability by running a second independent search in MinAtar SpaceInvaders. The 6 evolved RA functions display similar benefits of providing informative gradients in the log-ratio range [−1,0], and rapid saturation beyond that range, matching the analysis in Section 6.3. A separate run in MinAtar Breakout similarly ranked DAIL among the top-5 evolved functions, confirming both the robustness of the evolutionary procedure and DAIL’s effectiveness.
>
>
> ### **Table 2:  DAIL generalizes across different discriminator regularization strategies better than GAIL!**
> **Summary**: We assess DAIL’s generalization performance under various discriminator regularization strategies, including weight decay, an additional entropy bonus, and spectral normalization of the discriminator weights. We also consider the case without any regularization. DAIL outperforms GAIL across 3/5 regularization regimes, while only slightly lagging behind on the other two. This shows  that DAIL better complements different discriminator regularization strategies used in AIL.
>
> |Algo|Env|none|w-decay|entropy|spectral|grad-pen|
> |---|---|----|-------|-------|--------|--------|
> |DAIL|Asterix|0.88±0.03|1.33±0.03|0.12±0.01|0.92±0.03|0.66±0.03|
> ||Breakout|0.81±0.07|0.74±0.08|0.91±0.02|0.77±0.07|1.01±0.00|
> ||SpaceInvaders|0.71±0.07|0.81±0.01|0.80±0.01|0.70±0.09|0.90±0.00|
> ||**Overall**|0.80±0.03|**0.96±0.03**|0.61±0.01|**0.80±0.04**|**0.85±0.01**|
> |GAIL|Asterix|1.18±0.03|1.44±0.03|0.48±0.03|0.22±0.03|0.52±0.04|
> ||Breakout|0.76±0.07|0.52±0.10|0.89±0.01|0.33±0.10|0.85±0.07|
> ||SpaceInvaders|0.61±0.09|0.34±0.09|0.81±0.00|0.42±0.08|0.81±0.03|
> ||**Overall**|**0.85±0.04**|0.76±0.04|**0.73±0.01**|0.32±0.05|0.73±0.03|
>
> ### **Table 10: DAIL outperforms OpenAI Evolution Strategies!**
> **Summary**: OpenAI Evolution Strategies (Open-ES) is a black-box meta-learning algorithm known for its strong performance. Still, DAIL strongly outperforms it on most environments.
> |Env|DAIL|OPEN-ES|
> |---|----|-------|
> |SpaceInvaders|**0.90±0.00**|0.73±0.05|
> |Asterix|0.66±0.03|**1.27±0.04**|
> |Breakout|**1.01±0.00**|0.38±0.10|
> |HalfCheetah|**0.90±0.00**|0.62±0.02|
> |Walker2d|**0.95±0.00**|0.73±0.02|
> |Hopper|**1.65±0.01**|1.09±0.03|
> |Reacher|0.63±0.01|**0.83±0.01**|
> |Ant|**0.85±0.02**|0.51±0.03|
>
> ### **Table 11:  DAIL outperforms additional f-divergences and Wasserstein AIL baselines!**
>
> **Summary**: DAIL strongly outperforms other baselines on 5 out of 7 baselines
> | Environment | DAIL | Pearson | Sq-Hellinger | TV | WGAIL |
> |-------------|------|---------|--------------|----|-------|
> | Asterix     | **0.66±0.03** | -0.03±0.00 | -0.02±0.00 | -0.01±0.00 | 0.52±0.04 |
> | Breakout    | **1.01±0.00** | -0.01±0.00 | -0.01±0.00 | -0.01±0.00 | 0.91±0.06 |
> | Ant         | **0.85±0.02** | 0.57±0.02 | 0.82±0.03 | 0.19±0.05 | 0.80±0.02 |
> | HalfCheetah | **0.90±0.00** | 0.49±0.04 | 0.86±0.01 | -0.01±0.01 | 0.87±0.01 |
> | Hopper      | **1.65±0.01** | 0.90±0.11 | 1.49±0.04 | 1.59±0.04 | 1.50±0.01 |
> | Reacher     | 0.63±0.01 | 0.88±0.05 | **0.94±0.01** | 0.86±0.07 | 0.42±0.01 |
> | Walker2d    | 0.95±0.00 | 0.54±0.05 | **0.98±0.00** | 0.91±0.01 | 0.89±0.01 |

---

### Comment · Area_Chair_qi7e · 2025-11-26

Dear Reviewers,

Thank you for your time and effort in reviewing the submission. A reminder that the author–reviewer discussion period is about to conclude in one week. If you have not already done so, please review the authors’ rebuttals and engage in the discussion with the authors. Thanks!

Best,
Your AC

---

### Author Response · Authors · 2025-11-30
**Summary Following the OpenReview Leak**

It is unfortunate that the OpenReview leak prematurely ended the discussion phase, especially before reviewers had the chance to engage with our rebuttals. Given this situation, we expect that the final decision will rely more heavily on the AC/SAC/PC. To help mitigate the resulting bottleneck and support an informed assessment, we provide a concise summary below.

### Key contributions
This work makes the following key contributions to adversarial imitation learning (AIL):

- introduces the **first framework** that uses meta-learning to discover reward-assignment (RA) functions in AIL
- demonstrates that the discovered algorithm (DAIL) **generalizes** across a wide range of unseen environments, policy optimizers and discriminator regularizers, highlighting its robustness, broad applicability, and future relevance
- provides a **thorough analysis** of its performance, offering insight into why the meta-learned RA function is both desirable and effective

Together, these contributions represent a substantive advancement in the field of imitation learning.

### ⁠Positive Reviews
**3/4 reviewers** rated our paper positively (scores 6–8) prior to the rebuttal, more specifically:
- Reviewer **5SFA** and **6x5G** noted that our paper is the first to meta-learn RA functions in AIL
- Reviewer **5SFA** praised the presentation quality of our paper
- Reviewer **6x5G** praised the soundness of our approach, and the extensive analysis and ablations we conducted
- Reviewers **5SFA**, **6x5G**, and **GF4H** praised the strength of our experiment results, and that our method outperforms current baselines significantly

### Rebuttal
Unfortunately, we did not have the opportunity to engage directly with Reviewer **JUMG** (score 2). Nonetheless, we are confident that our rebuttal, supported by new experimental results, fully addresses the reviewer’s main concerns.

We hope this summary aids the decision process, and we sincerely appreciate your hard work.

---

### Meta-Review · Area_Chair_qAgm · 2026-01-04

**Summary:**

Across the four reviews, the overall assessment is mixed but leans positive: three reviewers rate the paper above the acceptance threshold (scores 6-8), while one reviewer (JUMG) is strongly negative (score 2) and raises substantial concerns about novelty, theory, and experimental design. The paper proposes DAIL, a meta-learning framework for discovering reward-assignment (RA) functions in adversarial imitation learning (AIL) using an LLM-guided evolutionary search. Positive reviewers emphasize that RA function design is underexplored in AIL, that the meta-learning framing is novel in this context, that the discovered function is concise and easy to integrate, and that empirical results across Brax and MinAtar show consistent improvements over standard AIL baselines (e.g., GAIL/AIRL/FAIRL). The main concerns informing the decision centered on (i) limited theoretical grounding or guarantees (including that the discovered RA function does not correspond to a known f-divergence and lacks convergence guarantees), (ii) the strength of the generalization claim given that meta-training is conducted on a single environment (MinAtar SpaceInvaders) and tested on a small set of unseen environments, (iii) whether the gains should be attributed to RA design versus the broader adversarial optimization dynamics, and (iv) whether the search process and comparisons sufficiently isolate the effect of the LLM-guided evolutionary procedure (including comparisons to alternative search/meta-learning baselines and other AIL variants such as Wasserstein-based methods and additional divergences). The rebuttal adds several experiments and clarifications (additional baselines including WGAIL and other divergences, stability evidence via an independent evolutionary run, a search on MinAtar Breakout where DAIL appears among top candidates, comparisons to OpenAI Evolution Strategies under matched budget, discriminator-regularization generalization, clarified compute cost, and clarifications around static vs. dynamic RA), which address many empirical and methodological questions, but the lack of formal guarantees and the breadth of generalization evidence remain partially unresolved.

**Reviewer Concerns:**

Concerns addressed by the rebuttal:
1. Additional baselines beyond GAIL/AIRL (5SFa, JUMG): The authors add comparisons to additional f-divergence RA functions and to a Wasserstein-based AIL baseline (WGAIL), and report that DAIL outperforms these baselines across the evaluated environments (Table 11; also referenced as updates to Table 3 / Appendix I).
2. Generalization across discriminator regularization regimes (6x5G, JUMG): The rebuttal reports results under multiple discriminator regularizers (none, weight decay, entropy bonus, spectral normalization, gradient penalty), showing DAIL outperforming GAIL in 3/5 regimes and being close in the others (Table 2).
3. Stability and reproducibility of the evolutionary procedure (6x5G): The authors report an additional independent evolutionary run on MinAtar SpaceInvaders yielding top functions with similar structural behavior, and additionally report that an evolutionary run on MinAtar Breakout ranks DAIL among the top-5 evolved functions (Table 8 / Figure 8; Section 6.4 / Appendix F). They also state that performance differences among top-k candidates are not statistically significant due to similar functional forms.
4. Effect of LLM-guided search vs simpler meta-optimization (JUMG): The rebuttal includes a matched-budget comparison against OpenAI Evolution Strategies (200 inner-loop evaluations) and reports that Open-ES underperforms DAIL overall, with qualitative commentary that Open-ES discovers irregular/non-smooth functions (Table 10; Appendix H).
5. Why Wasserstein is used as the fitness objective and relation to Wasserstein-based imitation (JUMG): The authors justify Wasserstein as a stable evaluation signal (metric, symmetric, no support-overlap requirement), argue direct optimization is computationally expensive, clarify that density-ratio formulations aid cross-environment applicability, and note that they already use a 1-Lipschitz discriminator via gradient penalty; they also describe WGAIL and state it underperforms DAIL in their experiments.
6. Compute cost and practicality of meta-training (6x5G): The authors clarify the sampling budget (20 samples/generation × 10 generations = 200 samples) and report wall-clock and GPU-hour cost for meta-training on SpaceInvaders (≈3 hours wall-clock; 6 GPU-hours using 2 GPUs), noting that Table 7 training times include parallel training of 16 agents enabled by a JAX pipeline.
7. IQ-Learn underperformance and Wasserstein cost matrix (6x5G): The rebuttal states they used IQ-Learn’s official implementation with default hyperparameters and notes similar results reported elsewhere (Appendix J). They also specify that Euclidean cost is used for Wasserstein distance.
8. Clarification on “static vs dynamic” RA (GF4H): The authors clarify that DAIL’s RA is fixed during each inner-loop AIL run (meta-testing), while RA functions evolve across outer-loop meta-training; “dynamic RA” is discussed only as future work.

Concerns partially addressed or still outstanding:
1. Theoretical guarantees / formal grounding (6x5G, GF4H, JUMG): While the authors provide intuition (monotonic reward in log-ratio) and argue that meta-learning is often empirically driven, they do not provide convergence guarantees or formal conditions under which DAIL should succeed/fail. The critique that DAIL does not correspond to a known divergence (and thus lacks standard guarantees) remains outstanding.
2. Strength of the generalization claim given meta-training scope (6x5G, JUMG): The rebuttal provides additional evidence (independent run on SpaceInvaders; a run on Breakout where DAIL appears among top-5), but the request to demonstrate consistency of discovered RA functions across multiple distinct meta-training environments (e.g., search on Brax environments like Ant/Walker, then test elsewhere) is not directly satisfied. The evidence remains limited to MinAtar meta-training environments as reported here.
3. Novelty framing vs prior RA analyses and the source of stability in AIL (JUMG, GF4H): The authors add missing citations and argue that RA choice affects stability and that their contribution is meta-learning RA beyond human-designed divergences, but the disagreement about whether RA changes can “fundamentally” stabilize adversarial optimization versus addressing discriminator/optimization dynamics remains a conceptual point not conclusively resolved by the rebuttal.

**Reviewer Scores:**

Reviewer 5SFa: Likely unchanged. The reviewer is strongly positive; the rebuttal adds requested baselines (additional divergences / WGAIL), which primarily reinforce their confidence rather than changing the overall stance.

Reviewer 6x5G: Likely unchanged or slightly higher. This reviewer’s main concerns relate to the lack of theoretical guarantees and insufficient generalization evidence; the rebuttal adds stability evidence (independent evolutionary run), additional meta-training evidence (Breakout run), regularization generalization, compute clarification, and more baselines, which address several experimental questions, though the theoretical gap remains.

Reviewer GF4H: Likely unchanged or slightly higher. The rebuttal clarifies the static/dynamic RA confusion and reiterates the authors’ stance on novelty and theory; however, the reviewer’s concern about limited theoretical rigor is not fully resolved.

Reviewer JUMG: Likely to increase modestly but remain negative. The rebuttal directly addresses multiple critiques by adding missing citations, adding comparisons to WGAIL and additional divergences, presenting a matched-budget comparison to OpenAI ES, reporting results under multiple discriminator regularizers, clarifying compute cost and implementation details (including IQ-Learn and the Wasserstein cost matrix), and clarifying that MinAtar is visually based. However, the reviewer’s core objections, limited theoretical novelty/guarantees and concerns that generalization evidence is insufficient without broader multi-environment meta-training, appear only partially addressed, so a substantial score change is unlikely based on the information provided here.

---

### Decision · Program_Chairs · 2026-01-26

Accept (Poster)